# Regulation of gliotoxin biosynthesis and protection in *Aspergillus* species

**Patrícia Alves de Castro**[1], **Ana Cristina Colabardini**[1], **Maísa Moraes**[1], **Maria Augusta Crivelente Horta**[1], **Sonja L. Knowles**[2], **Huzefa A. Raja**[2], **Nicholas H. Oberlies**[2], **Yasuji Koyama**[3], **Masahiro Ogawa**[3], **Katsuya Gomi**[4], **Jacob L. Steenwyk**[5], **Antonis Rokas**[5], **Relber A. Gonçales**[6,7], **Cláudio Duarte-Oliveira**[6,7], **Agostinho Carvalho**[6,7], **Laure N. A. Ries**[8]\*, **Gustavo H. Goldman**[1]\*

**1** Faculdade de Ciências Farmacêuticas de Ribeirão Preto, Universidade de São Paulo, Ribeirão Preto, Brazil, **2** Department of Chemistry and Biochemistry, University of North Carolina at Greensboro, North Carolina United States of America, **3** Noda Institute for Scientific Research, 338 Noda, Chiba, Japan, **4** Department of Bioindustrial Informatics and Genomics, Graduate School of Agricultural Science, Tohoku University, Sendai, Japan, **5** Department of Biological Sciences, Vanderbilt University, Nashville, Tennessee, United States of America, **6** Life and Health Sciences Research Institute (ICVS), School of Medicine, University of Minho, Braga, Portugal, **7** ICVS/3B's—PT Government Associate Laboratory, Guimarães/Braga, Portugal, **8** MRC Centre for Medical Mycology at the University of Exeter, Geoffrey Pope Building, Exeter, United Kingdom

\* rieslaure13@gmail.com (LNAR); ggoldman@usp.br (GHG)

**Data Availability Statement:** All relevant data are within the manuscript and its Supporting Information files. The BioProject ID in the NCBI's

## Abstract

*Aspergillus fumigatus* causes a range of human and animal diseases collectively known as aspergillosis. *A. fumigatus* possesses and expresses a range of genetic determinants of virulence, which facilitate colonisation and disease progression, including the secretion of mycotoxins. Gliotoxin (GT) is the best studied *A. fumigatus* mycotoxin with a wide range of known toxic effects that impair human immune cell function. GT is also highly toxic to *A. fumigatus* and this fungus has evolved self-protection mechanisms that include (i) the GT efflux pump GliA, (ii) the GT neutralising enzyme GliT, and (iii) the negative regulation of GT biosynthesis by the *bis*-thiomethyltransferase GtmA. The transcription factor (TF) RglT is the main regulator of GliT and this GT protection mechanism also occurs in the non-GT producing fungus *A. nidulans*. However, the *A. nidulans* genome does not encode GtmA and GliA. This work aimed at analysing the transcriptional response to exogenous GT in *A. fumigatus* and *A. nidulans*, two distantly related *Aspergillus* species, and to identify additional components required for GT protection. RNA-sequencing shows a highly different transcriptional response to exogenous GT with the RglT-dependent regulon also significantly differing between *A. fumigatus* and *A. nidulans*. However, we were able to observe homologs whose expression pattern was similar in both species (43 RglT-independent and 11 RglT-dependent). Based on this approach, we identified a novel RglT-dependent methyltransferase, MtrA, involved in GT protection. Taking into consideration the occurrence of RglT-independent modulated genes, we screened an *A. fumigatus* deletion library of 484 transcription factors (TFs) for sensitivity to GT and identified 15 TFs important for GT self-protection. Of these, the TF KojR, which is essential for kojic acid biosynthesis in *Aspergillus oryzae*, was also essential for virulence and GT biosynthesis in *A. fumigatus*, and for GT protection in *A. fumigatus*, *A. nidulans*, and *A. oryzae*. KojR regulates *rglT*, *gliT*, *gliJ* expression and sulfur

BioProject database is PRJNA729661. https://www.ncbi.nlm.nih.gov/bioproject/PRJNA729661.

**Funding:** São Paulo Research Foundation (FAPESP) grant numbers 2016/12948-7 (PAC), 2017/14159-2 (LNAR), and 2018/10962-8 (GHG) and 2016/07870-9 (GHG), and Conselho Nacional de Desenvolvimento Científico e Tecnológico (CNPq) 301058/2019-9 and 404735/2018-5 (GHG), both from Brazil. J.L.S. and A.R. are supported by the Howard Hughes Medical Institute through the James H. Gilliam Fellowships for Advanced Study program. AR's laboratory received additional support from a Discovery grant from Vanderbilt University, the Burroughs Wellcome Fund, the National Science Foundation (DEB-1442113), and the National Institutes of Health/National Institute of Allergy and Infectious Diseases (R56AI146096 and R01AI153356). S.L.K. is supported by the National Institutes of Health via the National Center for Complementary and Integrative Health (F31 AT010558). A.C., R.A.G., and C.D.-O. were supported by the Fundação para a Ciência e a Tecnologia (FCT) (PTDC/MED-GEN/28778/2017, UIDB/50026/2020, and UIDP/50026/2020), the Northern Portugal Regional Operational Program (NORTE 2020), under the Portugal 2020 Partnership Agreement, through the European Regional Development Fund (ERDF) (NORTE-01-0145-FEDER-000039), the ICVS Scientific Microscopy Platform, member of the national infrastructure PPBI - Portuguese Platform of Bioimaging (PPBI-POCI-01-0145-FEDER-022122), the European Union's Horizon 2020 research and innovation program under grant agreement 847507, and the "la Caixa" Foundation (ID 100010434) and FCT under the agreement LCF/PR/HP17/52190003. Individual support was provided by FCT (SFRH/BD/141127/2018 to C.D.-O.). We thank Universidade do Minho, Portugal and Universidade do São Paulo, Brazil for providing support for the scientific collaboration (G.H.G. and A.C., Edital USP-UMinho 2019. The funders had no role in study design, data collection and analysis, decision to publish, or preparation of the manuscript.

**Competing interests:** The authors have declared that no competing interests exist.

metabolism in *Aspergillus* species. Together, this study identified conserved components required for GT protection in *Aspergillus* species.

## Author summary

*A. fumigatus* secretes mycotoxins that are essential for its virulence and pathogenicity. Gliotoxin (GT) is a sulfur-containing mycotoxin, which is known to impair several aspects of the human immune response. GT is also toxic to different fungal species, which have evolved several GT protection strategies. To further decipher these responses, we used transcriptional profiling aiming to compare the response to GT in the GT producer *A. fumigatus* and the GT non-producer *A. nidulans*. This analysis allowed us to identify additional genes with a potential role in GT protection. We also identified 15 transcription factors (TFs) encoded in the *A. fumigatus* genome that are important for conferring resistance to exogenous gliotoxin. One of these TFs, KojR, which is essential for *A. oryzae* kojic acid production, is also important for virulence in *A. fumigatus* and GT protection in *A. fumigatus*, *A. nidulans* and *A. oryzae*. KojR regulates the expression of genes important for gliotoxin biosynthesis and protection, and sulfur metabolism. Together, this work identified conserved components required for gliotoxin protection in *Aspergillus* species.

## Introduction

*Aspergillus fumigatus* is a saprophytic fungus that can cause a variety of human and animal diseases known as aspergillosis [1]. The most severe of these diseases is invasive pulmonary aspergillosis (IPA), a life-threatening infection in immunosuppressed patients [2,3]. *A. fumigatus* pathogenicity is a multifactorial trait that depends in part on several virulence factors, such as thermotolerance, growth in the presence of hypoxic conditions, evasion and modulation of the human immune system and metabolic flexibility [3–7]. Another important *A. fumigatus* virulence attribute is the production of secondary metabolites (SMs). The genes that encode SMs are generally organized in biosynthetic gene clusters (BGCs) [8] and *A. fumigatus* has at least 598 SM-associated genes distributed among 33 BGCs [9,10]. SMs can cause damage to the host immune system, protect the fungus from host immune cells, or can mediate the acquisition of essential nutrients [11–18]. Gliotoxin (GT) is the best studied *A. fumigatus* SM and has been detected *in vivo* in murine models of invasive aspergillosis (IA), in human cancer patients [19], and in isolates derived from patients with COVID-19 and aspergillosis secondary co-infections [20].

Numerous modes of action for GT in the mammalian host have been described: (i) GT interferes with macrophage-mediated phagocytosis through prevention of integrin activation and actin dynamics, resulting in macrophage membrane retraction and failure to phagocytose pathogen targets [21]; (ii) GT inhibits the production of pro-inflammatory cytokines secreted by macrophages and the activation of the NFkB regulatory complex [22]; (iii) GT interferes with the correct assembly of NADPH oxidase through preventing p47phox phosphorylation and cytoskeletal incorporation as well as membrane translocation of subunits p47phox, p67phox and p40phox [23]; and (iv) GT inhibits neutrophil chemoattraction by targeting the activity of leukotriene A4 (LTA4) hydrolase, an enzyme that participates in LTA biosynthesis [24].

GT is a sulfur-containing mycotoxin, a member of the epipolythiopiperazines, produced by different fungal species, including *Gliocadium fimbriatum* (from which it was originally isolated and named accordingly), *A. fumigatus* and closely related non-pathogenic species [25,26], and also by species of *Trichoderma* and *Penicillium* [27–30]. In *A. fumigatus*, a BGC on chromosome VI contains 13 *gli* genes responsible for GT biosynthesis and secretion [30]. GT biosynthesis is tightly regulated because it interferes with and depends on several cellular pathways that regulate sulfur metabolism [cysteine (Cys) and methionine (Met)], oxidative stress defenses [glutathione (GSH) and ergothioneine (EGT)], methylation [S-adenosyl-methionine (SAM)], and iron metabolism (Fe–S clusters) [31–35]. Regulation of GT biosynthesis involves numerous transcription factors (TFs), protein kinases, transcriptional and developmental regulators, regulators of G-protein signalling as well as chromatin modifying enzymes [30].

Even though the regulation of GT biosynthesis is well characterized, regulation of endogenous protection from GT—which is also highly toxic to the fungus—is less well understood. *A. fumigatus* self-protection against GT is predicted to be based on the following mechanisms: (i) the major facilitator superfamily transporter GliA, which is part of the GT BGC, catalyses GT efflux; (ii) the reversible enzymatic activity of the oxidoreductase GliT, and (iii) the negative regulation of GT biosynthesis through the off switch mechanism of the *S*-adenosylmethionine-dependent gliotoxin *bis*-thiomethyltransferase GtmA. GliA is responsible for transferring GT out of fungal cells and deletion of *gliA* increases susceptibility to GT [36]. GliT produces the final toxic form of GT by catalysing disulphide bridge closure of the precursor dithiol gliotoxin (dtGT) [30,37]. However, if there is excess GT production, GliT is able to reduce GSH and produce dtGT attenuating GT toxicity [30,37]. GtmA, whose gene is not located in the GT BGC, is able to convert dtGT into bisdethiobis(methylthio)-gliotoxin (bmGT) and to attenuate GT production postbiosynthetically [38–41]. It is thought that the primary role of GtmA is a decrease in GT biosynthesis and not a back up for GliT and toxin neutralisation [30].

Until recently, the TF regulating *gliT* has remained elusive. It has previously been shown that *gliT* is not regulated by GliZ, a Zn(II)$_2$Cys$_6$ TF that is part of the gliotoxin BGC and is required for gliotoxin biosynthesis [37]. The TF RglT was shown to regulate not only GliZ but several other *gli* genes, including *gliT* [30,36,37,42], through directly binding to the respective promoter regions during GT-producing conditions. Interestingly, RglT and GliT were shown to have similar roles in GT protection in *A. nidulans*, a fungus that does not produce GT [42]. The genome of *A. nidulans* lacks homologs for *gliA* and *gtmA*. Therefore, the aim of this work is to elucidate additional components required for GT protection in *Aspergillus* species by comparing the transcriptional response of *A. fumigatus* and *A. nidulans* to exogenous GT. By using transcriptional profiling (RNA-seq), we investigate the influence of RglT in GT protection in the GT-producer *A. fumigatus* and in the GT-non producer *A. nidulans*. Although they are both members of the genus *Aspergillus*, the two species are distantly related and the sequence divergence of their genome sequences is on par to that of the human and fish genomes [43,44]. We identified several novel genetic determinants dependent or not on RglT and that could be involved in GT protection, including a novel methyltransferase encoded by *mtrA* whose deletion confers GT-sensitivity in both *A. fumigatus* and *A. nidulans*. We also screened an *A. fumigatus* deletion library of 484 null mutants and identified, in addition to RglT, 15 TFs that are potentially important for GT self-protection. We found that one of these TFs is a KojR ortholog, previously reported as regulator of the kojic acid production gene cluster in *A. oryzae*, is important as well for *A. nidulans* and *A. oryzae* GT protection and involved in *A. fumigatus* virulence, GT self-protection and GT and bmGT biosynthesis. KojR regulates the inorganic sulfur assimilation and transssulfuration pathways when *A. fumigatus* is exposed to GT.

## Results

### A. fumigatus RglT controls the expression of GT- and other SM-encoding genes when exposed to exogenous GT

To determine which genes are under the transcriptional control of RglT in the presence of exogenous GT, we performed RNA-sequencing (RNA-seq) of the *A. fumigatus* wild-type (WT) and Δ*rglT* strains when exposed to 5 μg/ml GT for 3 h. In these conditions, *A. fumigatus* is protecting itself from the effects of GT, as has previously been shown [45]. Differentially expressed genes (DEGs) were defined as those with a minimum log2 fold change of 2 [log2FC $\geq$ 1.0 and $\leq$ -1.0; *p*-value < 0.05; FDR (false discovery rate) of 0.05]. DEG comparisons were carried out for: (i) the WT strain, comparing the GT condition to the control (GT-free condition) to determine which genes are modulated after 3 h exposure to GT; and (ii) when comparing the *rglT* deletion strain to the WT strain in the presence of GT to determine which genes are under the regulatory control of RglT in these conditions.

In the WT strain, 260 genes were down-regulated and 270 genes up-regulated when comparing the GT to the control condition (S1 Table). FunCat (Functional Categorisation) (https://elbe.hki-jena.de/fungifun/fungifun.php) analysis for the WT strain comparisons showed a transcriptional up-regulation of genes coding for proteins involved in non-vesicular ER transport, translation, mitochondrion, respiration and electron transport (*p*-value < 0.01; Fig 1A). FunCat analysis for the down-regulated genes showed enrichment for genes involved in homeostasis of metal ions, detoxification involving cytochrome P450, C-compound and carbohydrate metabolism, and secondary metabolism (*p*-value < 0.01; Fig 1A). When comparing the WT to the Δ*rglT* strain in the presence of GT, a total of 269 genes were down-regulated and 694 genes were up-regulated (S2 Table). FunCat enrichment analysis of these DEGs showed a transcriptional up-regulation of genes encoding proteins involved in secondary metabolism, C-compound and carbohydrate metabolism, cellular import, and siderophore-iron transport (*p*-value < 0.01; Fig 1B). FunCat analysis of down-regulated DEGs showed enrichment of transport facilities, lipid, fatty acid and isoprenoid metabolism, and homeostasis of cations (*p*-value < 0.01; Fig 1B). These results suggest that *A. fumigatus* adapts to the prolonged presence of GT through transcriptionally inducing cellular homeostasis and detoxification processes and that RglT is important for these processes.

More specifically, RglT is important for the expression of 11 out of the 13 genes present in the GT BGC (including *gliT* and *gliA*) as well as *gtmA* (which is not part of the GT BGC) in the presence of exogenous GT (Fig 1C). Interestingly, *gliI* and *gliJ* expression are not altered by the presence of GT in the WT and in the Δ*rglT* strains (Fig 1C). RNA-seq analysis also revealed transcriptional modulation of other SM-encoding genes. For example, BGC genes encoding components for the biosynthesis of pyripyropene, naphthopyrone, fumagillin, fumiquinazolines, and fumigaclavine were shown to be under the transcriptional control of RglT, with this TF repressing these genes under prolonged GT exposing conditions (Fig 1D).

These results suggest that RglT is important not only for GT biosynthesis and self-protection, but also for the regulation of the expression of genes involved in the production of other SMs.

### Comparison of the A. fumigatus and A. nidulans transcriptional responses when exposed to GT

Next, we performed RNA-seq for the *A. nidulans* WT (the GT MIC for *A. nidulans* is 30 μg/ml) and Δ*rglT* strains in the same conditions (5 μg/ml GT for 3 h) as described above and identified DEGs. The same two comparisons were carried out as described above for *A. fumigatus*

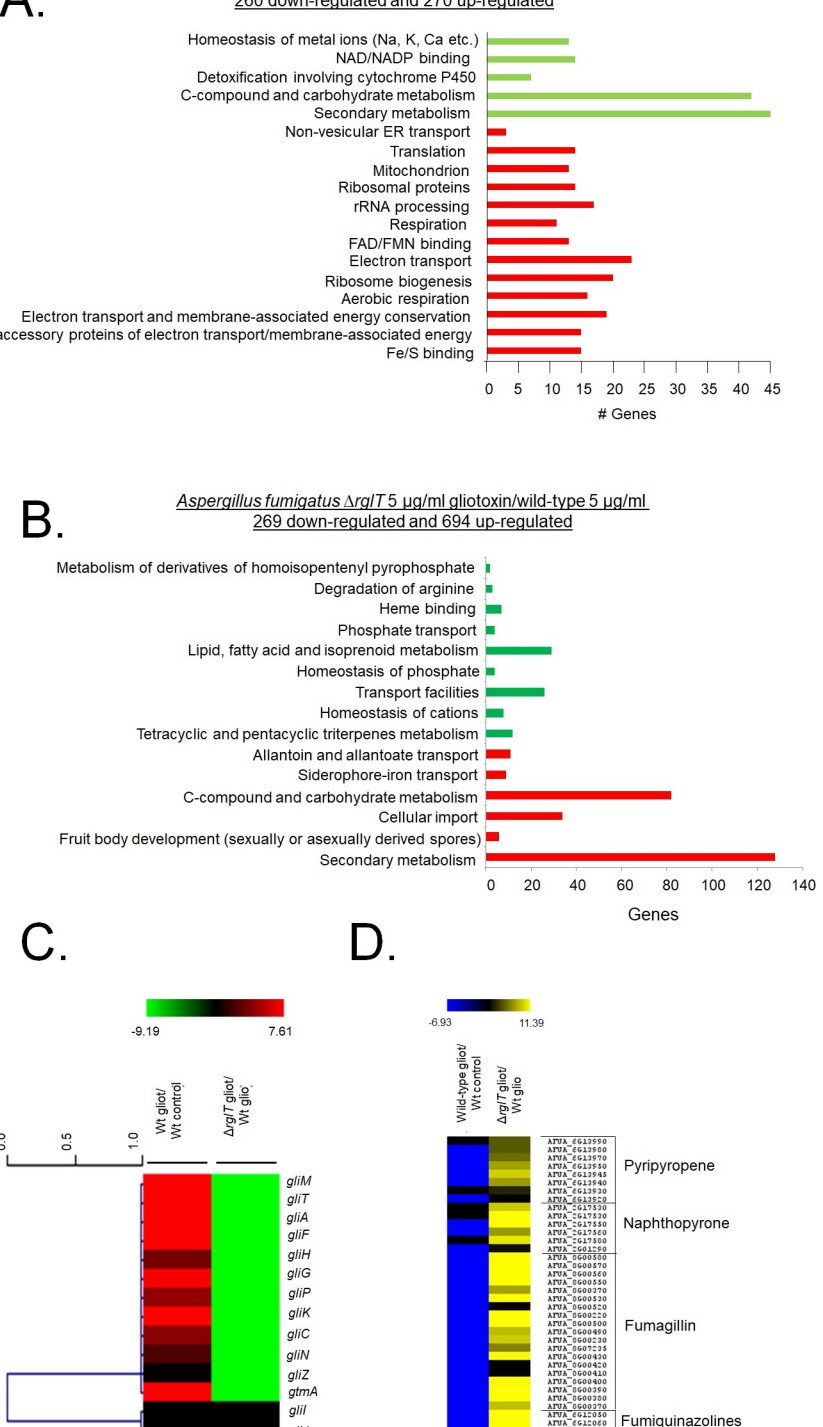

**Fig 1. Functional characterisation (FunCat) of significantly differently expressed genes (DEGs) identified by RNA-sequencing in the *A. fumigatus* Δ*rglT* strain.** (A) FunCat analysis of DEGs up-regulated in the *A. fumigatus* wild-type and in (B) Δ*rglT* strain in comparison to the wild-type (WT) strains when exposed to 5 μg/ml GT for 3 h. (C) Heat map depicting the log2 fold change (Log2FC) of differentially expressed genes (DEGs), as determined by RNA-sequencing, and encoding enzymes present in the GT BGC required for GT biosynthesis. The gene *gtmA* was also included in this heat map. (D) Heat map depicting the Log2FC of differentially expressed genes (DEGs), as determined by RNA-sequencing, and encoding enzymes required for secondary metabolite (SM) biosynthesis. In both (C) and (D)

log2FC values are based on the wild-type strain exposed to GT in comparison to the wild-type control and the Δ*rglT* exposed to GT in comparison to the wild-type exposed to GT. Heat map scale and gene identities are indicated. Hierarchical clustering was performed in MeV (http://mev.tm4.org/), using Pearson correlation with complete linkage clustering.

(WT GT vs. WT Control and Δ*rglT* GT vs WT GT). In the *A. nidulans* WT strain, 678 genes were down-regulated and 675 genes were up-regulated when comparing the GT to the control condition (S3 Table). We were not able to observe any Funcat enrichment for these DEGs and then we employed gene ontology (GO; https://elbe.hki-jena.de/fungifun/fungifun.php) enrichment analyses for the WT strain that showed transcriptional up-regulation of genes coding for proteins involved in mitochondrial function, such as ATP synthesis coupled proton transport and cytochrome-c oxidase activity, as well as ergot alkaloid biosynthetic process and a heterogeneous set of cellular components (*p*-value < 0.01; Fig 2A). GO analysis for the down-regulated genes showed enrichment for a heterogeneous set of genes involved in biological processes and cellular components (*p*-value < 0.01; Fig 2A). When comparing the WT to the Δ*rglT* strain in the presence of GT, 132 genes were down-regulated and 68 genes were up-regulated (S4 Table). GO enrichment analyses of the Δ*rglT* strain showed a transcriptional up-regulation of genes encoding proteins involved in nucleotide binding and cellular components (*p*-value < 0.01; Fig 2B). GO analysis of down-regulated DEGs in the Δ*rglT* strain showed enrichment of flavin adenine dinucleotide binding and oxidation-reduction processes (*p*-value < 0.01; Fig 2B). In contrast to *A. fumigatus*, where prolonged exposure to GT resulted in 530 identified DEG in the WT strain (about 5.3% of the *A. fumigatus* 9,929 protein coding sequences, CDSs), in *A. nidulans* 1,353 DEG (about 13.5% of the *A. nidulans* 9,956 CDSs) were identified in the WT strain in these conditions. Similarly, in *A. fumigatus*, RglT controls the expression of 963 genes (about 9.7% of the CDSs) when exposed to GT for 3 h, whereas the number of genes regulated by RglT in *A. nidulans* is smaller (200 genes, about 2% of the CDSs). Together, these results suggest very significant differences in the transcriptional response to GT and for the role of RglT in mediating this response between *Aspergilus* species.

To determine whether a conserved transcriptional response to GT in both *A. fumigatus* and *A. nidulans* exists, we searched for homologous genes whose expression pattern was similar (i) between the WT strains and (ii) RglT-dependent in both fungal species in the presence of GT. When *A. fumigatus* wild-type DEGs were compared to *A. nidulans* wild-type DEGs, 35 and 11 homologs were up- and down-regulated in both species, respectively (S3 Table). Furthermore, a total of 11 genes were found: six genes were dependent on RglT for expression [GliT (AN3963/AFUA_6G09740), a methyltransferase (AN3717/AFUA_6G12780), GprM (AN3567/AFUA_7G05300), a Major Facilitator Superfamily transporter (AN1472/AFUA_8G04630), AN8167/AFUA_5G02950, and AN1470/AFUA_5G8G04260 with unknown functions]; and the other five genes were dependent on RglT for repression [lysozyme activity (AN6470/AFUA_6G10130), L-PSP endoribonuclease (AN5543/AFUA_5G033780), oxidoreductase activity (AN9051/AFUA_7G00700), NmrA-transcription factor (AN9531/AFUA_7G06920), and an ATP-binding cassette transporter (AN7879/AFUA_1G10390)] (Fig 2C and S4 Table).

We observed 43 homologs that were RglT-independent (S4 Table) and among them, we identified as induced by GT a putative cysteine synthase (AN7113/AFUA_4G03930) and ZrcA, a zinc/cadmium resistance protein (AN10876/AFUA_7G06570) (S4 Table). Another RglT-independent gene is the GliT paralogue AN6963/AFUA_5G03540; interestingly, this gene is upregulated in *A. nidulans* and downregulated in *A. fumigatus* upon GT exposure (S4 Table).

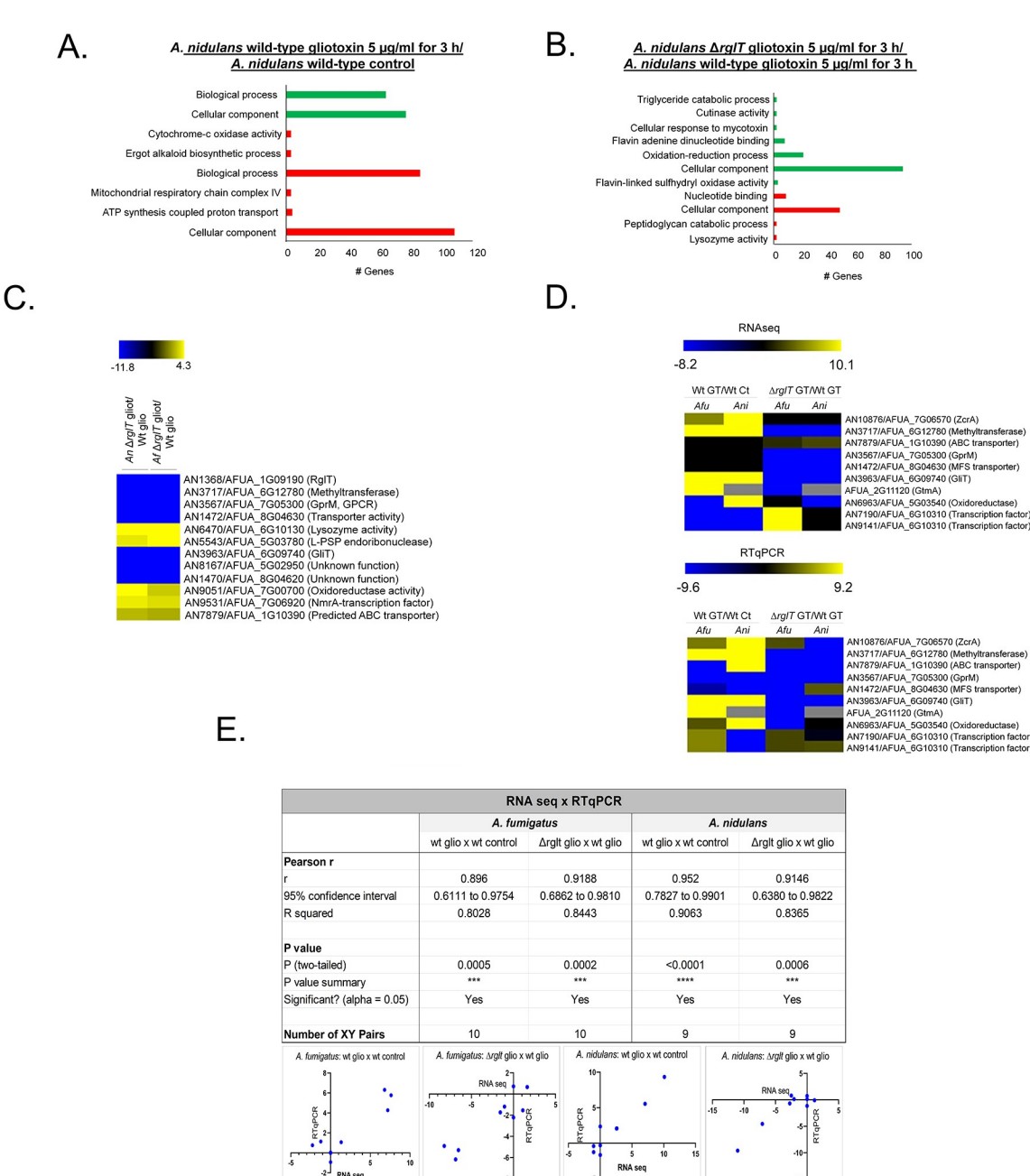

**Fig 2. Functional characterisation (FunCat) of significantly differently expressed genes (DEGs) identified by RNA-sequencing in the *A. nidulans* Δ*rglT* strain.** (A) FunCat analysis of DEGs up-regulated in the *A. nidulans* wild-type in the presence of GT when compared to the control condition, and in (B) the Δ*rglT* strain in comparison to the wild-type (WT) strain when exposed to 5 μg/ml GT for 3 h. (C) Heat map depicting the log2 fold change (Log2FC) of differentially expressed homologous genes (DEHGs), as determined by RNA-sequencing, that have their expression dependent on RglT. (D) Upper panel: heat map depicting the log2 fold change (Log2FC) of 9 DEHGs and *gtmA* as determined by RNA-sequencing. Lower panel: heat map depicting the log2 fold change (Log2FC) of 9 DEHGs and *gtmA* as determined by RT-qPCR. Log2FC values are based on the *A. fumigatus* (*Afu*) or *A. nidulans* (*Ani*) wild-type strains exposed to GT when compared to the *Afu* or *Ani* wild-type control and the *Afu* or *Ani* Δ*rglT* strains exposed to GT when compared to the *Afu* or *Ani* wild-type exposed to GT. Heat map scale and gene identities are indicated. Hierarchical clustering was performed in MeV (http://mev.tm4.org/), using Pearson correlation with complete linkage clustering. For the RT-qPCR experiments, the values represent the average of three independent biological repetitions (each with 2 technical repetitions). (D) Pearson correlation analysis between the expression of 10 genes as measured by RT-qPCR and RNA-sequencing. A positive correlation was seen for all analysed datasets.

RNA-seq results were confirmed by RT-qPCR for the majority of the 9 selected *A. fumiga-tus*/*A. nidulans* homologs and for the *A. fumigatus* GtmA-encoding gene (*A. nidulans* lacks a homolog) (Figs 2D and 2E and S1). The expression of these 10 genes showed a high level of correlation with the RNA-seq data (Pearson correlation from 0.896 to 0.952; Fig 2E).

Taken together, our results suggest that the transcriptional responses of *A. fumigatus* and *A. nidulans* to GT are very different, although we were able to observe 54 homologs whose transcriptional response was similar in both species. However, we note that we used an equal exposure to GT (5 μg/ml GT for 3 hours) for both species but *A. nidulans* shows greater susceptibility to GT than *A. fumigatus*.

## A novel methyltransferase (MtrA) is important for GT self-defense in *Aspergilli*

Among the genes dependent on *Aspergilli* RglT, we identified a methyltransferase (named MtrA, AN3717/AFUA_6G12780). (Fig 2C and S4 Table). We deleted this gene in both *A. nidu-lans* and *A. fumigatus* and tested the growth of two independent transformants for each deletion in the presence of GT (Figs 3A and S3). *A. fumigatus* ΔmtrA deletion strains showed about 20% growth reduction compared to the corresponding wild-type while *A. nidulans* ΔmtrA deletion strains presented about 60% growth reduction (Fig 3A). Fig 3B shows the taxonomic distribution of MtrA, RglT, GliT, and GT biosynthetic gene cluster (BGC) homologs across Eurotiomycetes and Sordariomycetes. Phylogenetically informed model testing of the distributions of RglT and MtrA suggests that the pattern of occurrence of RglT is statistically dependent on the distribution of MtrA but not vice-versa. This analysis suggests an evolutionary scenario in which the MtrA-based resistance mechanism is ancestral and the evolutionary recruitment of RglT regulation of GliT occurred subsequently.

These results strongly indicate that our approach to compare the transcriptomics of *A. fumigatus* and *A. nidulans* ΔrglT mutants was efficient to identify novel genes involved in GT-protection.

## Screening of A. *fumigatus* TF deletion strains for sensitivity to GT

Our RNA-seq results show there are DEGs that are not dependent on RglT and suggest there additional TFs that could be important for GT protection in *Aspergilli*. To identify these possible TFs involved in the GT self-protection response, we took advantage of the existing *A. fumi-gatus* TF null deletion library [46] which we screened for sensitivity to 35 μg/ml of GT (*A. fumigatus* minimal inhibitory concentration for GT is 70 μg/ml). In addition to ΔrglT, 15 null mutants exhibited increased sensitivity to 35 μg/ml of GT (Figs 4 and 5). Only AFUA_3G09670 (*oefA*) is up-regulated in the ΔrglT mutant and only AFUA_4G13060 is up-regulated in the wild-type strain in our RNA-seq experiments (S1 and S2 Tables).

Previous studies have shown that GT biosynthesis and self-protection is intimately linked to sulfur metabolism, oxidative stress resistance, as well as iron and zinc metabolism [47–52]. To identify if any of the TFs identified in the screen are associated with these pathways, the null mutants were screened for growth in the presence of (i) methionine or cysteine as single sulfur sources (S2 Fig); (ii) oxidative stress-inducing agents such as allyl alcohol (converted to the oxidative stress-inducing compound acrolein by alcohol dehydrogenases), *t*-butyl hydroxyperoxide, menadione, and diamide (a GSH scavenger); and (iii) increased and depleted extracellular iron and zinc concentrations (Fig 6). None of these mutants showed altered growth compared to the WT strain in the presence of methionine or cysteine as single sulfur sources (S2 Fig). Six of the deletion strains [ΔrglT, ΔsreA (ΔAFUA_5G12060), ΔAFUA_5G140390, Δyap1 (ΔAFUA_6G09930), ΔAFUA_6G09870, and ΔAFUA_8G07360]

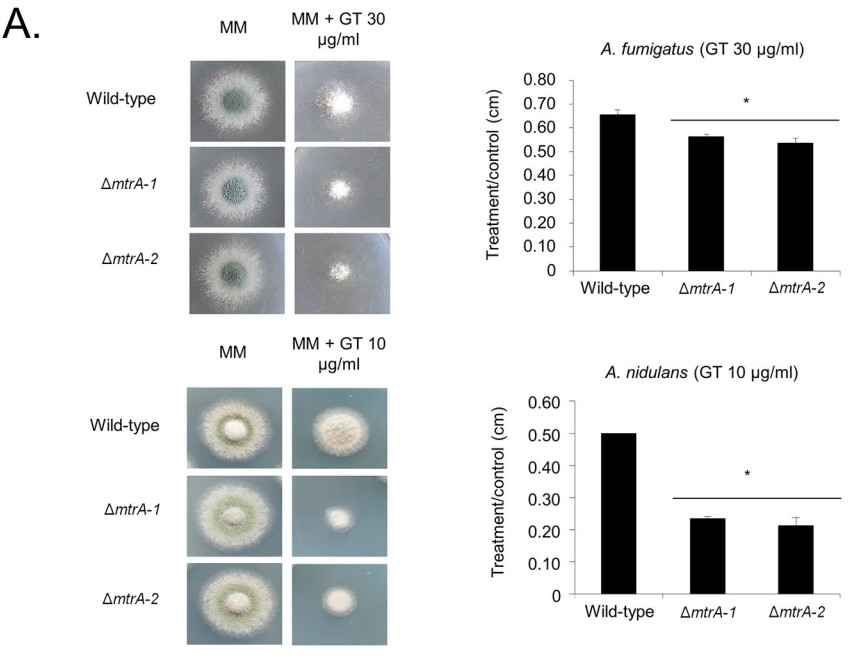

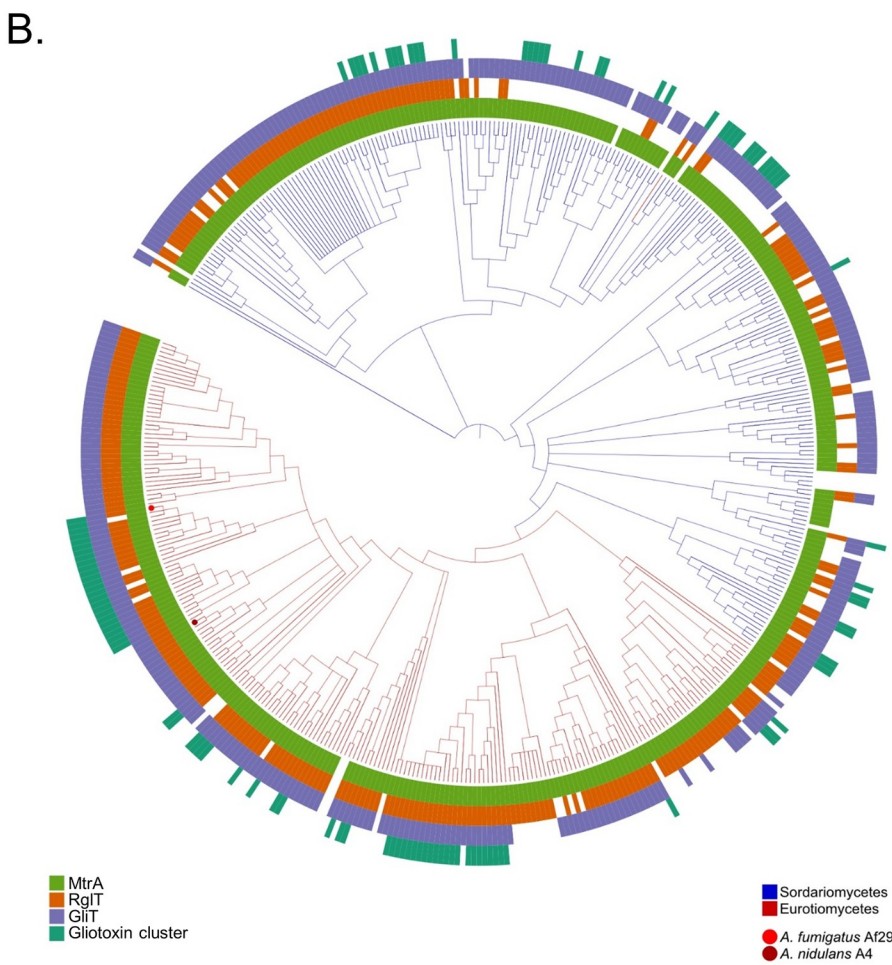

**Fig 3. The *A. fumigatus* and *A. nidulans* Δ*mtrA* mutants are more sensitive to GT.** (A) The *A. fumigatus* and *A. nidulans* wild-type and two independent Δ*mtrA* strains were grown for 48 hours at 37°C on MM in the absence or presence of 30 or 10 μg/ml of GT, respectively. The results are the average of three repetitions ± standard deviation. Statistical analysis was performed using a one-tailed, paired t-test when compared to the control condition (*, $p <$ 0.01). (B) The phylogenetic distribution of MtrA, RglT, GliT and GT biosynthetic gene cluster homologs across 458 fungal genomes. A 4-gene phylogeny of genomes from Eurotiomycetes (shown by the red branches) and Sordariomycetes (shown by the blue branches). For every tip in the phylogeny, the presence of MtrA, RglT, GliT and gliotoxin BGC homologs is depicted using light green, orange, blue, and dark green, respectively; absences are depicted in white. The dark green bar plots depict how many of the 13 Gli genes are present in the gliotoxin BGC homolog. The tip corresponding to the *A. fumigatus* Af293 genome is indicated by a red dot, and the tip corresponding to the *A. nidulans* A4 genome is indicated by a maroon dot.

were sensitive to at least one of the four oxidative stress-inducing agents (Fig 6A–6D). The ΔAFUA_8G07360 mutant is resistant to diamide (Fig 6D). Four strains (ΔAFUA_3G13920, ΔAFUA_5G11260, ΔAFUA_3G01100 and ΔAFUA_8G03970) had increased growth, while ΔAFUA_3G09670, ΔAFUA_5G06800, and ΔAFUA_5G12060 presented significantly decreased growth in iron starvation conditions (Fig 6F). Intriguingly, ΔAFUA_8G05460 had reduced growth upon zinc starvation but increased growth upon zinc excess (Fig 6G and 6H). The Δ*rglT* mutant has reduced growth in zinc starvation conditions but had increased growth in iron starvation conditions (Fig 6F and 6H).

Taken together, these results strongly suggest that some of the mutants identified as more sensitive to GT are also involved in pathways related to oxidative stress and iron and zinc metabolism.

## Production of gliotoxin and BmGT in the gliotoxin sensitive mutants

The Δ*rglT* and Δ*gliT* strains are not only sensitive to exogenous GT, but they also secrete significantly less GT [37,42]. To determine if the TF deletion strains from our screen are also impaired for GT and bmGT biosynthesis and secretion, we quantified extracellular GT and bmGT in culture supernatants of these strains when grown in GT-inducing conditions. As previously demonstrated [42], high performance liquid chromatography and mass spectrometry experiments showed that Δ*rglT* has reduced GT and produces a similar amount of bmGT when compared to the WT and *rglT* complemented strains (S5 Table, Fig 7A–7C). The Δ*kojR* (ΔAFUA_5G06800), ΔAFUA_5G12060, Δ*rgdA* (ΔAFUA_3G13920), ΔAFUA_8G07360, Δ*sreA* (ΔAFUA_5G11260), and ΔAFUA_2G17860 strains secrete significantly less GT when compared to the WT strain (Fig 7A and 7C). In contrast, the Δ*oefC* (ΔAFUA_3G09670), ΔAFUA_6G09870, and ΔAFUA_5G14390 strains secrete significantly more GT than the WT strain (Fig 7A and 7C). Production of bmGT is decreased in the Δ*kojR* (ΔAFUA_5G06800) but increased in most of the strains (10 out of 16), with the exception of Δ*rglT*, ΔAFUA_5G12060, Δ*rgdA* (ΔAFUA_3G13920), Δ*sreA* (ΔAFUA_5G11260), and ΔAFUA_2G17860 which produce similar concentrations of bmGT than the WT strain (Fig 7B and 7C). Interestingly, the Δ*kojR* (ΔAFUA_5G06800) strain is the only TF deletion strain with reduced GT and bmGT levels (Fig 7A–7C), whilst the ΔAFUA_8G03970, ΔAFUA_6G09930, and Δ*oefC* (ΔAFUA_3G09670) strains have increased levels of GT and bmGT in the supernatants (Fig 7A–7C).

Our results indicate that several TFs are involved in the induction or repression of GT and bmGT biosynthesis. In contrast, only for the Δ*kojR* strain are GT and bmGT extracellular levels significantly decreased, and we have chosen this TF for further characterisation.

## RglT and KojR are important for GT protection in *Aspergilli*

We have previously shown that RglT and GliT in *A. fumigatus* and *A. nidulans* have a similar role in protecting each fungus from the deleterious effects of GT [42]. Homologs of RglT and

| Gene ID | Structure | Family | Gliotoxin Inhibition of the null mutant (µg/ml) |
|---------|-----------|--------|--------------------------------------------------|
| RglT (AFUA_1G09190) | | GAL4-like Zn(II)2Cys6 | < 35 |
| AFUA_2G17860 | | GAL4-like Zn(II)2Cys6 | < 35 |
| RgdA (AFUA_3G13920) | | KilA-N domain | < 35 |
| OefC (AFUA_3G09670) | | GAL4-like Zn(II)2Cys6 | < 35 |
| KojR (AFUA_5G06800) | | GAL4-like Zn(II)2Cys6 | < 35 |
| SreA (AFUA_5G11260) | | GATA-type zinc finger | < 35 |
| AFUA_5G12060 | | GAL4-like Zn(II)2Cys6 | < 35 |
| AFUA_5G14390 | | GAL4-like Zn(II)2Cys6 | < 35 |
| AFUA_4G13060 | | ZnF_C2HC domain | < 35 |
| AFUA_6G09870 | | GAL4-like Zn(II)2Cys6 | < 35 |
| Yap1 (AFUA_6G09930) | | Basic-leucine zipper (bZIP) | < 35 |
| AFUA_8G07360 | | GAL4-like Zn(II)2Cys6 | < 35 |
| AFUA_8G07280 | | GAL4-like Zn(II)2Cys6 | < 35 |
| AFUA_8G05460 | | Basic-leucine zipper (bZIP) | < 35 |
| AFUA_8G03970 | | GAL4-like Zn(II)2Cys6 | < 35 |
| AFUA_3G01100 | | ZnF_C2HC domain | < 35 |

**Fig 4. List of the transcription factor mutants identified as more sensitive to gliotoxin (the domais organization are based on http://smart.embl-heidelberg.de/).**

GliT are present in a number of Eurotiomycetes and Sordariomycetes suggesting that this mechanism of GT protection is widespread, including in many species that are not GT producers [42]. Similarly to MtrA, phylogenetically informed model testing led to an evolutionary scenario in which the GliT-based resistance mechanism is ancestral and RglT-mediated regulation of GliT occurred subsequently [42]. To determine whether a phylogenetic relationship exists between RglT and KojR, we screened for the presence of GT and kojic acid BGCs as well

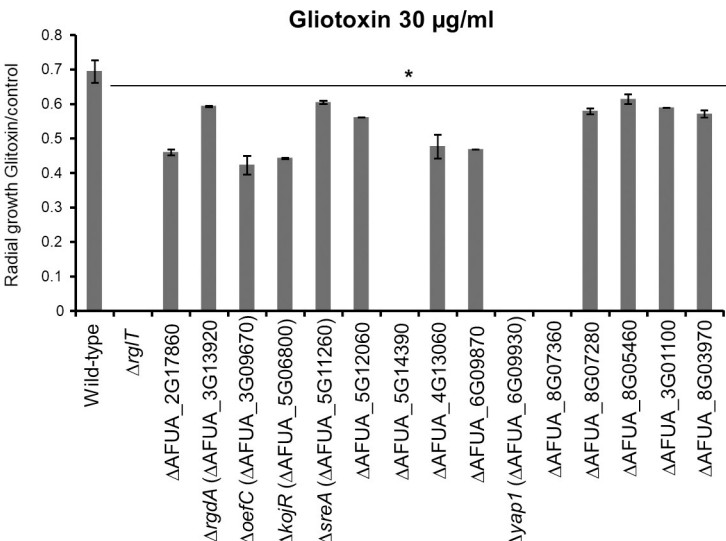

**Fig 5. Identification *A. fumigatus* TFs important for gliotoxin (GT) self-protection.** Strains were grown from $10^4$ conidia for 2 days at 37˚C on minimal medium (MM) supplemented with 30 μg/ml GT before colony radial diameter was measured. The results are expressed as the radial diameter of the treatment divided by the radial diameter of the growth in the control GT-free condition. The results are the means of three repetitions ± standard deviation. Statistical analysis was performed using a one-tailed, paired t-test when compared to the control condition (*, $p < 0.05$).

as RglT and KojR homologs across *Aspergillus* and *Penicillium* species and constructed a phylogenetic tree (Fig 8A). Results suggest that their origins predate the origin of both genera. In contrast, homologs of the kojic acid BGC are present in only *A. oryzae* and close relatives (Fig 8A). In contrast to RglT and GliT, examination of the phylogenetic distribution patterns of RglT and KojR revealed that they are not significantly correlated (scenario A). The same was true for KojR and the kojic acid BGC, suggesting that KojR was recruited specifically for kojic acid biosynthesis in *A. oryzae* and close relatives. In contrast, the distribution of the GT biosynthetic gene cluster is correlated with the presence of *RglT* (scenario B) [42], but not vice-versa; however, scenario A (lack of correlation between their distributions) also fit the observed data well (Table 1).

To determine whether KojR is also required for protection from GT in other fungi, the corresponding homologous genes were deleted in *A. nidulans* (chosen because it does not produce either GT or kojic acid) and *A. oryzae* (produces kojic acid). Similarly to *A. fumigatus*, deletion of *kojR* significantly increased sensitivity to exogenously added GT in both *A. nidulans* and *A. oryzae* (Fig 8B–8D). In contrast to deletion of *rglT*, which made all three species fully sensitive to GT, deletion of *kojR* resulted in only partial sensitivity to GT (Fig 8B–8D). We cannot explain the increased sensitivity to GT in *A. oryzae* Δ*kojR*, but these results suggest that RglT functions downstream of KojR and/or that RglT is regulated by KojR in these conditions. To address this, the *A. oryzae* WT, the *rglT* and the *kojR* deletion strains were first screened for kojic acid production, which can be seen by the intensity of the red color (formed by chelating ferric ions) on plates. As expected, deletion of *kojR* resulted in the absence of red colour and hence kojic acid production, whereas deletion of *rglT* did not affect kojic acid production in *A. oryzae* (Fig 8E). These results suggest that RglT does not regulate kojic acid production in *A. oryzae*.

## GliT and GtmA are partially dependent on *Aspergilli* KojR

To determine whether KojR is under the transcriptional control of RglT in the presence of GT, transcriptional levels of *kojR* in the *A. fumigatus* and *A. oryzae rglT* deletion strains were

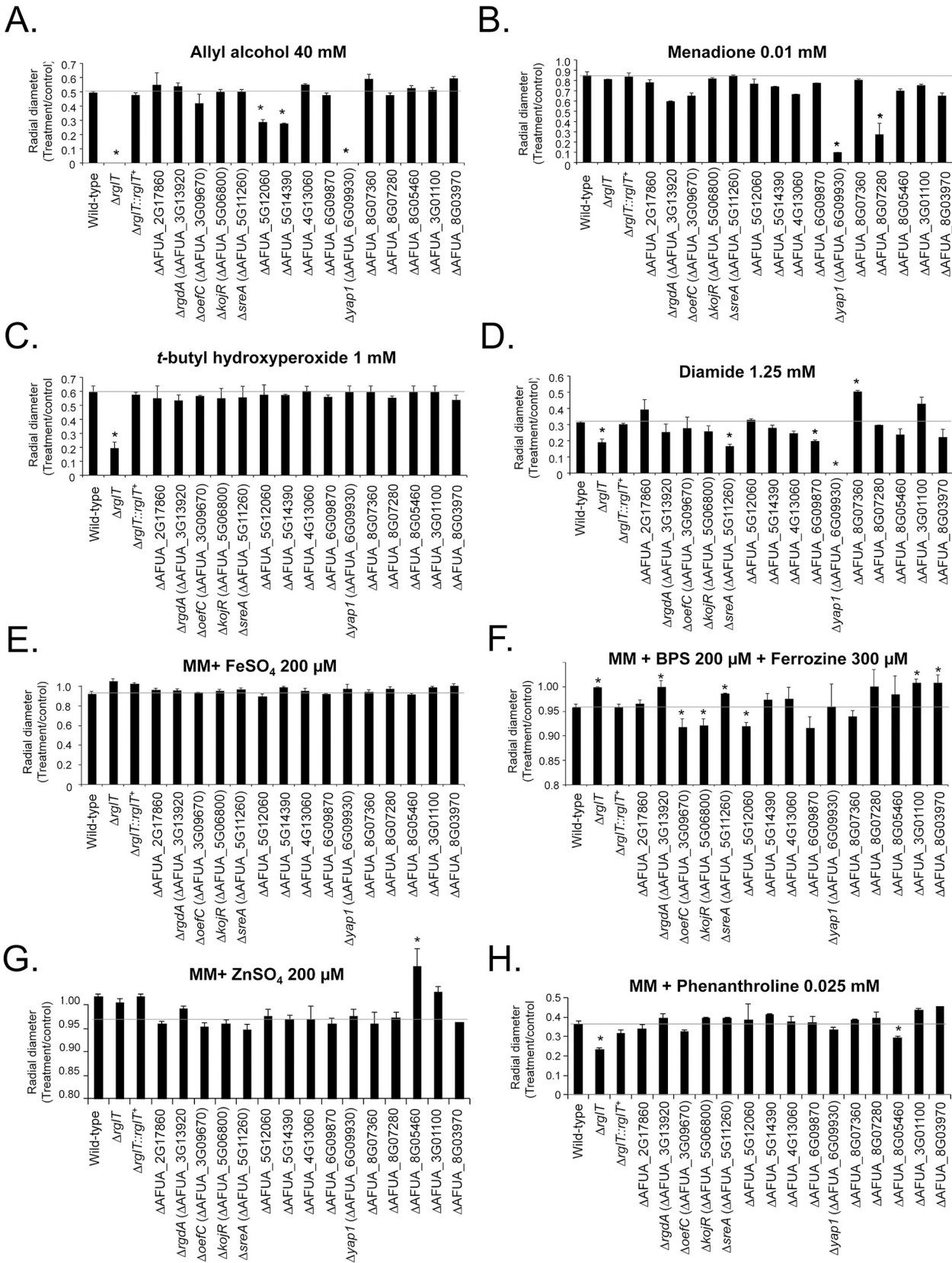

**Fig 6. Growth phenotypes of the *A. fumigatus* GT-sensitive transcription factor deletion strains.** Strains were grown from $10^4$ conidia for 5 days at 37°C on minimal medium supplemented with the indicated compounds. The results are expressed as the radial diameter of the treatment divided by the radial diameter of the growth in the control, drug-free condition. The results are the means of three repetitions ± standard deviation. Statistical analysis was performed using a one-tailed, paired t-test when compared to the control condition ($^*$, $p < 0.05$). (A) allyl alcohol 40 mM. (B) menadione 0.01 mM. (C) *t*-butyl hydroxyperoxide 1 mM. (D) diamide 1.25 mM. (E) FeSO$_4$ 200 μM. (F) BPS 200 μM+Ferrozine 300 μM. (G) ZnSO$_4$ 200 μM. (H) phenanthroline 0.025 mM.

measured by RT-qPCR (Fig 9A). In *A. fumigatus* and *A. oryzae*, deletion of *rglT* resulted in significantly decreased and increased *kojR* transcript levels, respectively, suggesting that RglT regulates *kojR* in these fungi (Fig 9A). To determine whether *rglT* is under the transcriptional

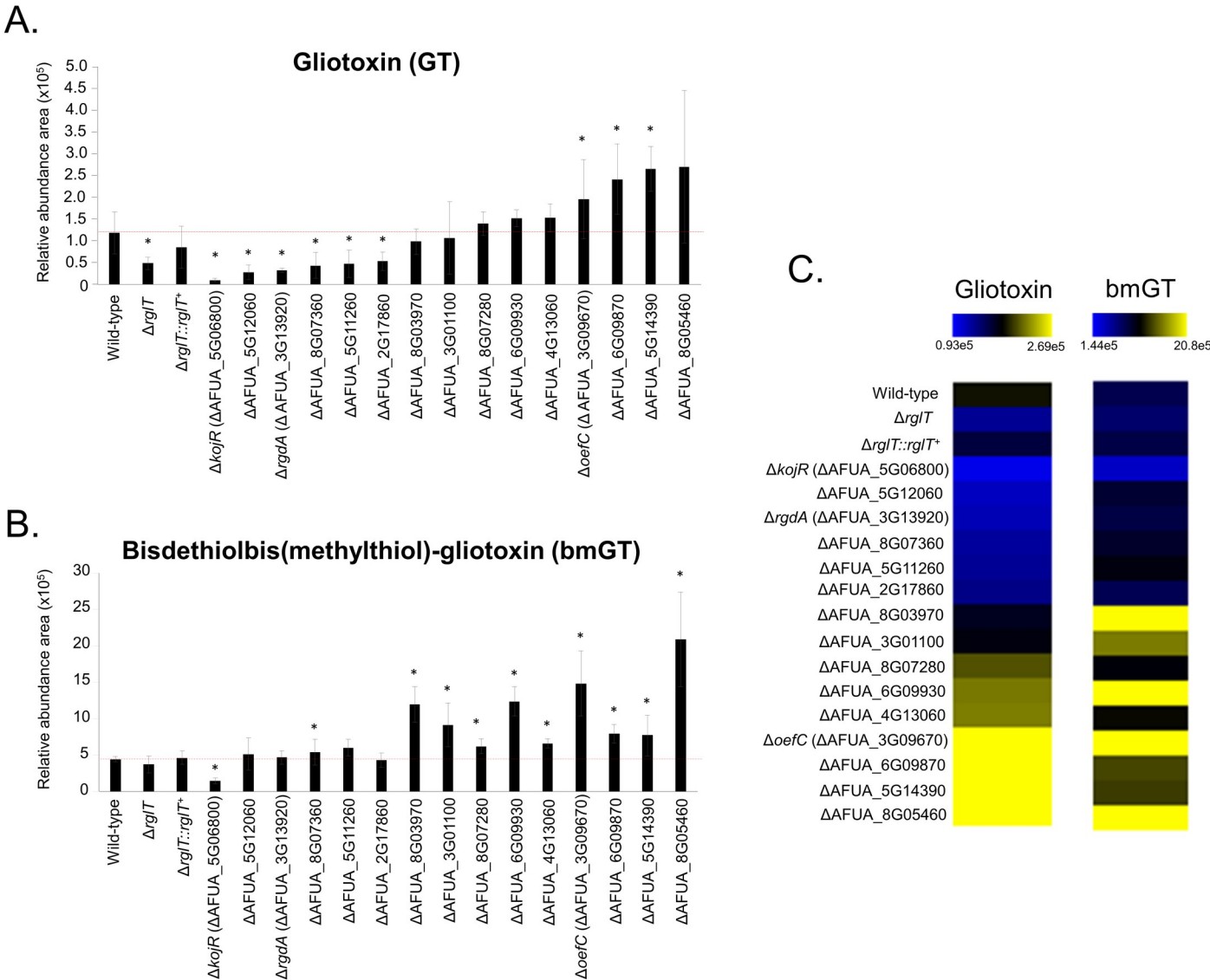

**Fig 7. Production of GT and bmGT in the *A. fumigatus* GT-sensitive TF deletion strains.** (A) Relative abundance of GT and (B) bisdethiobis(methylthio)-gliotoxin (bmGT) as measured by high performance liquid chromatography coupled to mass spectrometry. Results are the average of three repetitions ± standard deviation. Statistical analysis was performed using a one-tailed, paired t-test when comparing quantities to the relative abundance of the WT strain ($^*$, $p < 0.005$). (C) Heat map of the results from panels (A) and (B). Heat map scale and gene identities are indicated. Hierarchical clustering was performed in MeV (http://mev.tm4.org/), using Pearson correlation with complete linkage clustering.

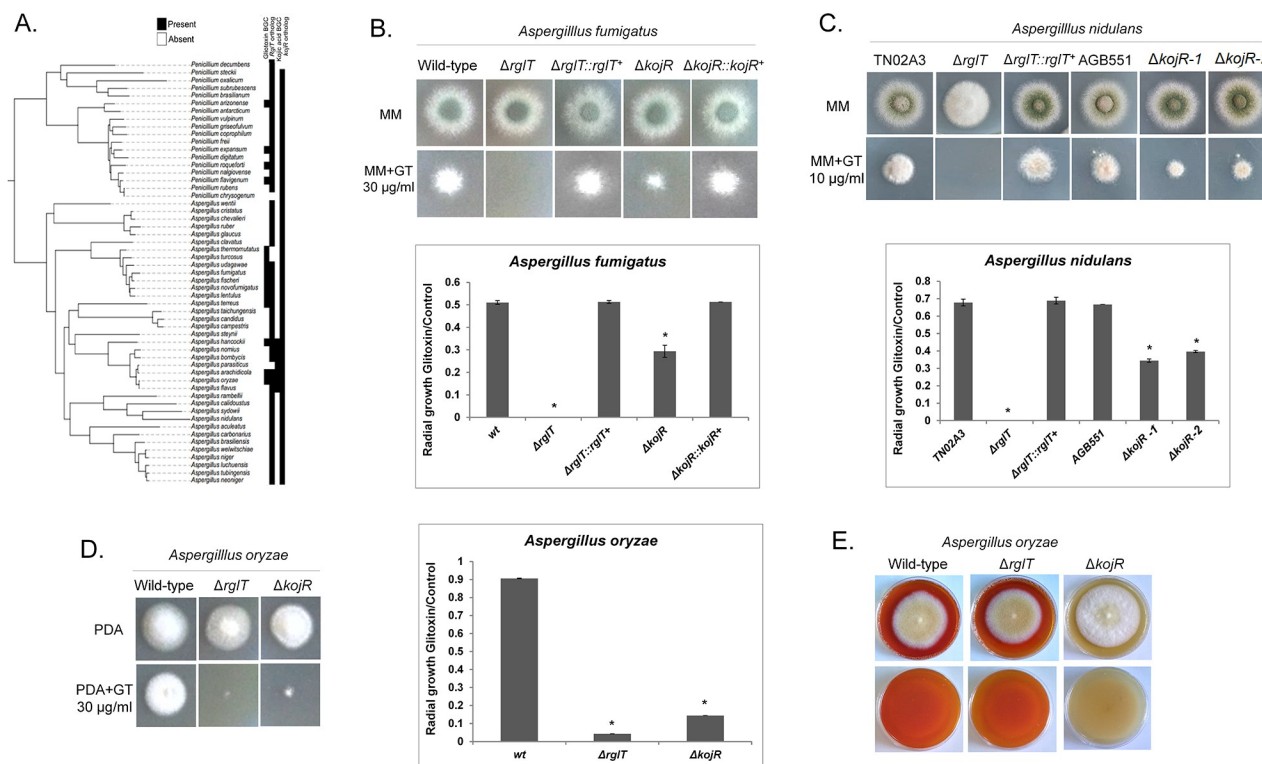

**Fig 8. *Aspergilli* RglT and KojR homologs are important for gliotoxin production and self-protection.** (A) Distribution of homologs of the gliotoxin and kojic acid biosynthetic gene clusters and of the RglT and KojR homologs amongst *Aspergillus* and *Penicillium* species. Homologs of the gliotoxin biosynthetic gene cluster are present across many (but not all) *Aspergillus* and *Penicillium* species, suggesting that the cluster was present in their common ancestor. Homologs of the kojic acid biosynthetic gene cluster are present only in *Aspergillus oryzae* and close relatives, suggesting that the cluster originated much more recently. Orthologs for RglT and KojR are broadly conserved in both genera, suggesting they were present in the *Aspergillus–Penicillium* common ancestor. Presence or absence of biosynthetic gene cluster homologs and of gene orthologs are depicted in black or white, respectively. (B-D) Pictures and graphs of the *A. fumigatus* (B), *A. nidulans* (C) and *A. oryzae* (D) wild-type, Δ*rglT*, and Δ*kojR* strains when grown for 48 h from $10^4$ conidia at 37°C on minimal medium (MM, *A. fumigatus*, *A. nidulans*) or PDA (*A. oryzae*) and MM/PDA+10 or 30 μg of gliotoxin. Graphs correspond to the radial diameter of the colony growth that is depicted in the pictures. Standard deviations represent the average of three biological replicates with $^*p < 0.05$ in a one-tailed, paired t-test. (E) Detection of kojic acid (KA) production in *A. oryzae* in medium containing 1 mM ferric ion (FeCl₃). A red colour indicates the presence of KA chelated with ferric ions. The first and second rows indicate the top and bottom of the plates, respectively.

control of KojR in the presence of GT, *rglT* transcript levels were quantified by RT-qPCR in the WT and *kojR* deletion strains (Fig 9B). In the *A. fumigatus* and *A. oryzae* Δ*kojR* strains, *rglT* transcript levels were significantly reduced, suggesting that KojR regulates *rglT* expression in the presence of exogenous GT (Fig 9B). We were unable to measure the expression levels of *A. nidulans kojR* and *rglT* in the absence or presence of GT because the PCR amplification

**Table 1. Akaike Information Criterium (AIC) of model fitting for phylogenetic correlation tests.**

| | Independent | Dependent X | Dependent Y | Interdependent |
|---|---|---|---|---|
| **X: *kojR*; Y: *rglT*** | <u>0.767</u> | 0.104 | 0.113 | 0.016 |
| **X: *kojR*; Y: kojic acid BGC** | <u>0.704</u> | 0.101 | 0.170 | 0.025 |
| **X: *rglT*; Y: gliotoxin BGC** | 0.336 | <u>0.471</u> | 0.068 | 0.125 |

BGC = biosynthetic gene cluster

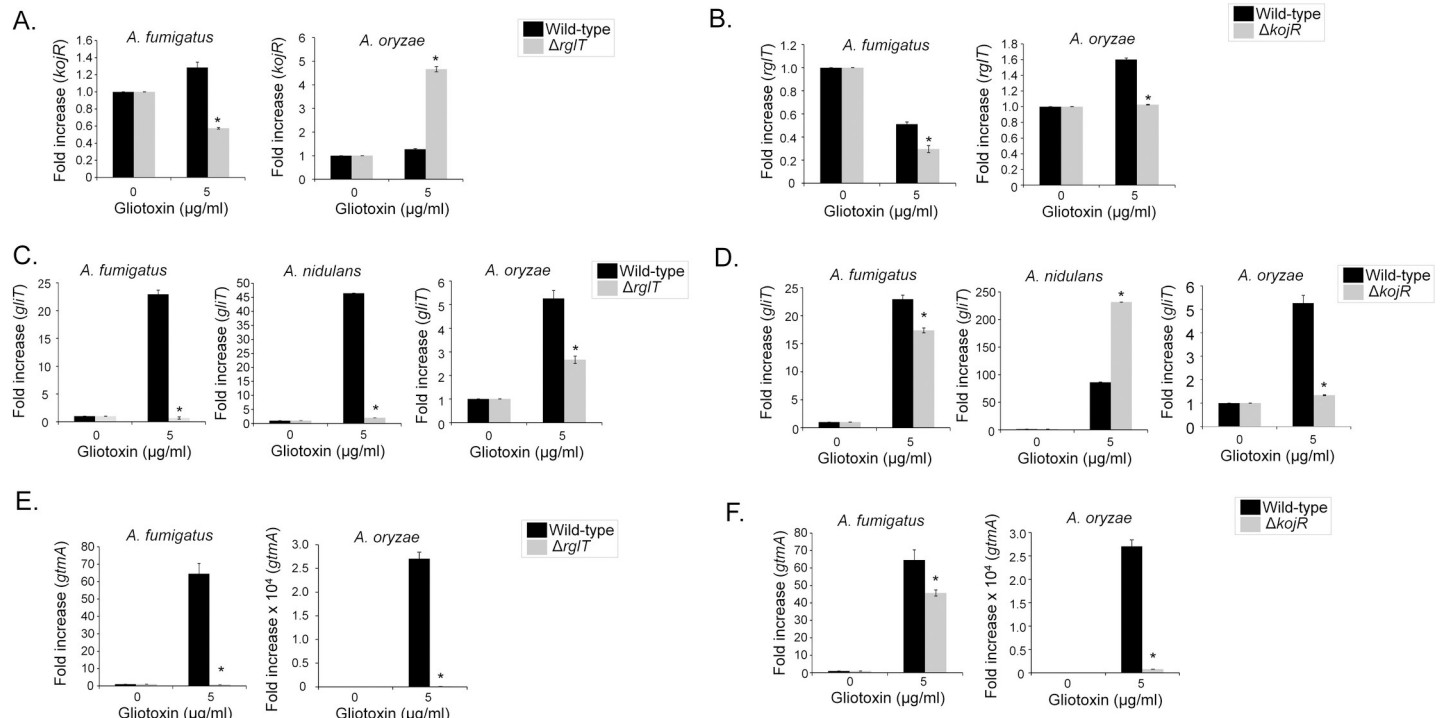

**Fig 9. GliT and GtmA are dependent on *Aspergilli* RglT and KojR.** *A. fumigatus*, *A. nidulans*, and *A. oryzae* strains were grown for 21 h at 37°C before 5 μg/ml of GT was added to cultures for 3 hours. RNA was extracted from all fungal strains, reverse transcribed to cDNA and RT-qPCRs were peformed. The expression of *gtmA*, *kojR* and *rglT* in the *A. nidulans* WT, Δ*rglT* and Δ*kojR* strains could not be measured due to either the absence of the gene (*gtmA*) or the absence of quantifyable mRNA. Results are the average of three biological replicates ± standard deviation. Statistical analysis was performed using a one-tailed, paired t-test when compared to the control condition (*, $p < 0.05$). (A-D) *A. fumigatus*, *A. nidulans*, and *A. oryzae* wild-type, Δ*rglT*, and Δ*kojR* strains were grown for 21 h at 37°C and mycelia was exposed to 5 μg/ml of GT for 3 hours. RT-qPCR analysis for *kojR* (A), *rglT* (B), *gliT* (C and E), and *rglT* (D and F) genes.

profiles were similar to the negative control (water), suggesting *kojR* and *rglT* have very low levels of expression in this fungus.

Finally, we further assessed the role of KojR in GT protection in *A. fumigatus*, *A. nidulans*, and *A. oryzae*. We were unable to detect any GT and bmGT production in *A. oryzae* wild-type, RglT and KojR (S4 Fig). We previously demonstrated that the expression of *gliT*, encoding an oxidoreductase with a GT neutralising function, is dependent on RglT in both *A. fumigatus* and *A. nidulans* when GT was added exogenously to the medium [42] (Figs 2C and 9C). In agreement with these data, the expression of *A. oryzae gliT* is decreased in the Δ*rglT* mutant, suggesting that *gliT* expression is dependent on RglT in all three *Aspergillus* species (Fig 9C). Furthermore, *gliT* expression was also dependent on KojR in these fungal species with KojR inducing *gliT* expression in *A. fumigatus* (about 20% reduction in the Δ*kojR* mutant) and *A. oryzae* (about a three-fold reduction in the Δ*kojR* mutant) and repressing *gliT* in *A. nidulans* (about 2.5-fold increase in the Δ*kojR* mutant; Fig 9D). Subsequently, *gtmA* transcript levels were also quantified as *gtmA* is also important for GT self-protection [38]. The expression of *gtmA* is also significantly decreased in *A. fumigatus* and *A. oryzae* Δ*rglT* mutants (Fig 9E) and about 20% and 25-fold, respectively, in the *A. fumigatus* and *A. oryzae* Δ*kojR* mutants (Fig 9F). No *gtmA* homolog is present in *A. nidulans*.

Together these results suggest: (i) a regulatory interplay exists between RglT and KojR in GT protection that differs between *Aspergillus* spp; (ii) RglT-dependent GliT protection from exogenous GT is a conserved mechanism in GT-producing and non-producing *Aspergillus* spp; (iii) the existence of further mechanisms of GT self-protection that significantly differ

between GT-producing and non-producing *Aspergillus spp*; and (iv) *A. fumigatus* KojR plays a role in the *gliT* and *gtmA* regulation in the presence of GT.

## *A. fumigatus* KojR is important for *gliJ* expression, inorganic sulfur assimilation, and virulence

We decided to investigate the influence of KojR on the expression of other genes that are important for GT production and protection. Fig 10 summarizes the contributions of some of these pathways and the different genes involved in GT biosynthesis and self-defense. We compared the mRNA accumulation of selected genes described in Fig 10 when the wild-type and the Δ*kojR* mutant were exposed to 5 μg/ml GT for 3 hours (Fig 11A). Thirteen *gli* genes in the GT BGC, and gtmA, were up-regulated when *A. fumigatus* wild-type is exposed to GT (Fig 11A). Interestingly, different from the RNA-seq data (Fig 1C), we were able to see increased *gliI* and *gliJ* mRNA levels when the wild-type is exposed to GT by using RT-qPCR (Fig 11A). These results emphasize that the *gli* genes are up-regulated when *A. fumigatus* is exposed to GT. The *gliP*, *gliG*, *gliF*, *gliH*, *gliN*, *gliA*, *gliT*, and *gtmA* (these two last ones previously shown in the Fig 9) have about 20 to 30% less mRNA accumulation in the Δ*kojR* mutant than in the wild-type while Δ*gliJ* has a two-fold increase in the wild-type and it is significantly down-regulated in the Δ*kojR* mutant (Fig 11A). The TF *gliZ* is about 10% more expressed in the Δ*kojR* mutant than in the wild-type while *gtmA* is about 20% less expressed in the Δ*kojR* mutant (Fig 11A).

Inorganic sulfur assimilation starts by the transport of sulfate by the sulfate transporter sB; through several steps of metabolic transformation, sulfate is subsequently converted into sulfide and then to methionine through the subsequent action of sC, sD, sF, CysB, MetB, CysD, and MetH (Figs 10 and 11B). In the *A. fumigatus* wild-type strain, there is an increased mRNA accumulation of all the genes of the inorganic sulfur assimilation (*sB*, *sC*, *sD*, *sF*, and *cysB*) including its transcriptional regulator *metR* and a decreased mRNA accumulation of selected genes in the transsulfuration pathway (*metB*, *metG*, *cysD*, *metH*, and *mecB*) upon GT exposure (Figs 10 and 11B). In contrast, the *sB* mRNA accumulation is significantly reduced about 10-fold in the Δ*kojR* mutant when compared to the wild-type strain (Fig 11B). There is a lower mRNA accumulation of *sB*, *sC*, *sD*, *sF*, *cysB*, and *cysD* when compared to the wild-type strain (Fig 11B). In contrast, *metB*, *metH*, and *mecB* have higher mRNA accumulation than the wild-type and there are no differences between both strains for *metG* (Fig 11B).

The increased expression of transsulfuration genes (*metB*, *metH*, and *mecB*) when the Δ*kojR* mutant is exposed to GT suggests that the Δ*kojR* mutant is trying to synthesize other sulfur metabolic intermediates through the transsulfuration pathway. We reasoned that if Δ*kojR* mutant has problems in inorganic sulfur assimilation, growth in the presence of organic sulfur sources, such as methionine or cysteine, could suppress the GT-sensitivity observed when Δ*kojR* is grown on MM+SO$_4$ as a single inorganic sulfur source (Figs 8B and 11C). The GT-sensitivity of the Δ*kojR* mutant is partially suppressed in MM+cysteine but not in MM+methionine (Fig 11C), strongly emphasizing not only the defects of assimilation of inorganic sulfur by Δ*kojR* in the presence of GT but also possible defects in methionine assimilation. Interestingly, the *A. nidulans* Δ*kojR* GT susceptibility is also suppressed by cysteine and only partially by methionine as a single sulfur source (Fig 11D) while cysteine (but not methionine) partially suppressed *A. oryzae* GT Δ*kojR* susceptibility (Fig 11E), strongly suggesting that *A. nidulans* and *A. oryzae* KojR homologs are also important for sulfur assimilation during GT protection. Additional evidence for that comes from the KojR-dependency on the expression *A. nidulans* and *A. oryzae* of *sB* sulfate transporter homologs upon GT exposure (Fig 11F). We also observed that GT-sensitivity of both *A. fumigatus* and *A. oryzae* Δ*rglT* mutants could not

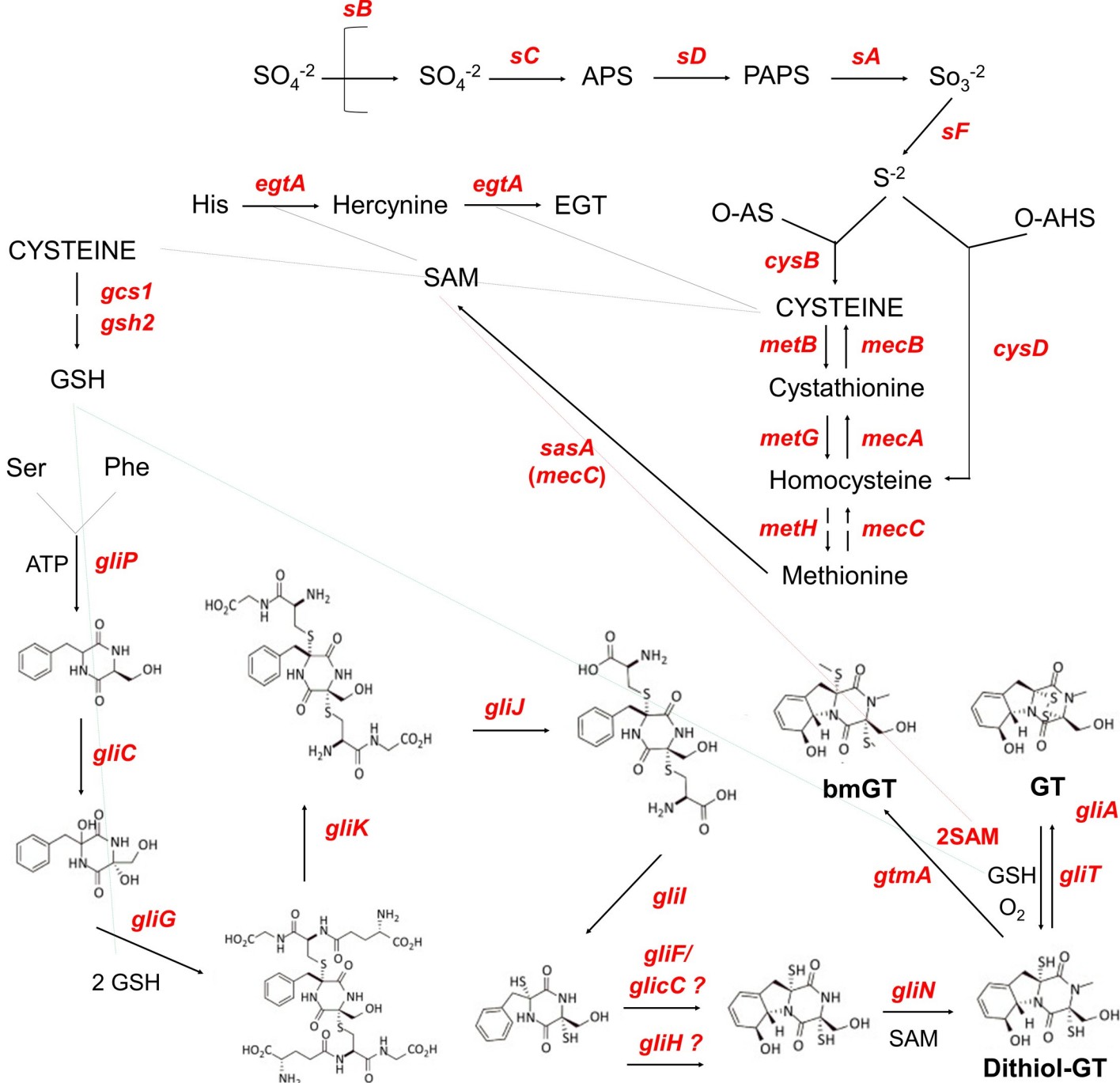

**Fig 10. Schematic representation of the pathways that are important for GT production and self-defense.** – The pathways of sulfate assimilation, transsulfuration, and GT production are shown in the scheme. The depicted genes are: *sB*, sulfate transporter; *sC*, ATP sulfurylase; *sD*, adenosine 5'-phosphosulfate (APS) kinase; *sA*, 3'-phosphoadenosine-5'-phosphosulfate (PAPS) reductase; *sF*, β-subunit of the sulfite reductase; AFUA_6G08920, α-subunit of the sulfite reductase; *cysB*, cysteine synthase; *metB*, cystathione-ϒ-synthase; *metG*, cysthationine-β-lyase; *metH*, methionine synthase; *mecC*, 5-adenosylmethionine synthetase; *mecA*, cystathionine-β-synthase; *mecB*, cystathionine- ϒ-lyase; *cysD*, homocysteine synthase; *gliP*, **n**on-ribosomal peptide synthetase; *gliC*, cytochrome P450 monooxygenase; *gliG*, glutathione-S-transferase; *gliK*, unknown protein; *gliJ*, membrane dipeptidase; *gliI*, 1-aminocyclopropane-1-carboxylic acid synthase; *gliF*, cytochrome P450 monooxygenase; *glicC*, cytochrome P450 monooxygenase; *gliH*, acetyl transferase; *gliN*, methyltransferase; *gliT*, gliotoxin sulfhydryl oxidase; *gliA*, major facilitator type glioxin transporter; *gcs1*, glutamate cysteine-ligase; *gsh2*, glutathione synthase; *egtA*, **ergothioneine synthase**; and *gtmA*, *bis*-thiomethyltransferase. **Abbreviations: O-AS,** *O*-acetylserine; **O-HS,** *O*-acetylhomoserine; **GSH, glutathione; SAM, S-adenosyl methionine; EGT, ergothioneine; bmGT,** bisdethiobis(methylthio)-gliotoxin; **and GT, gliotoxin.**

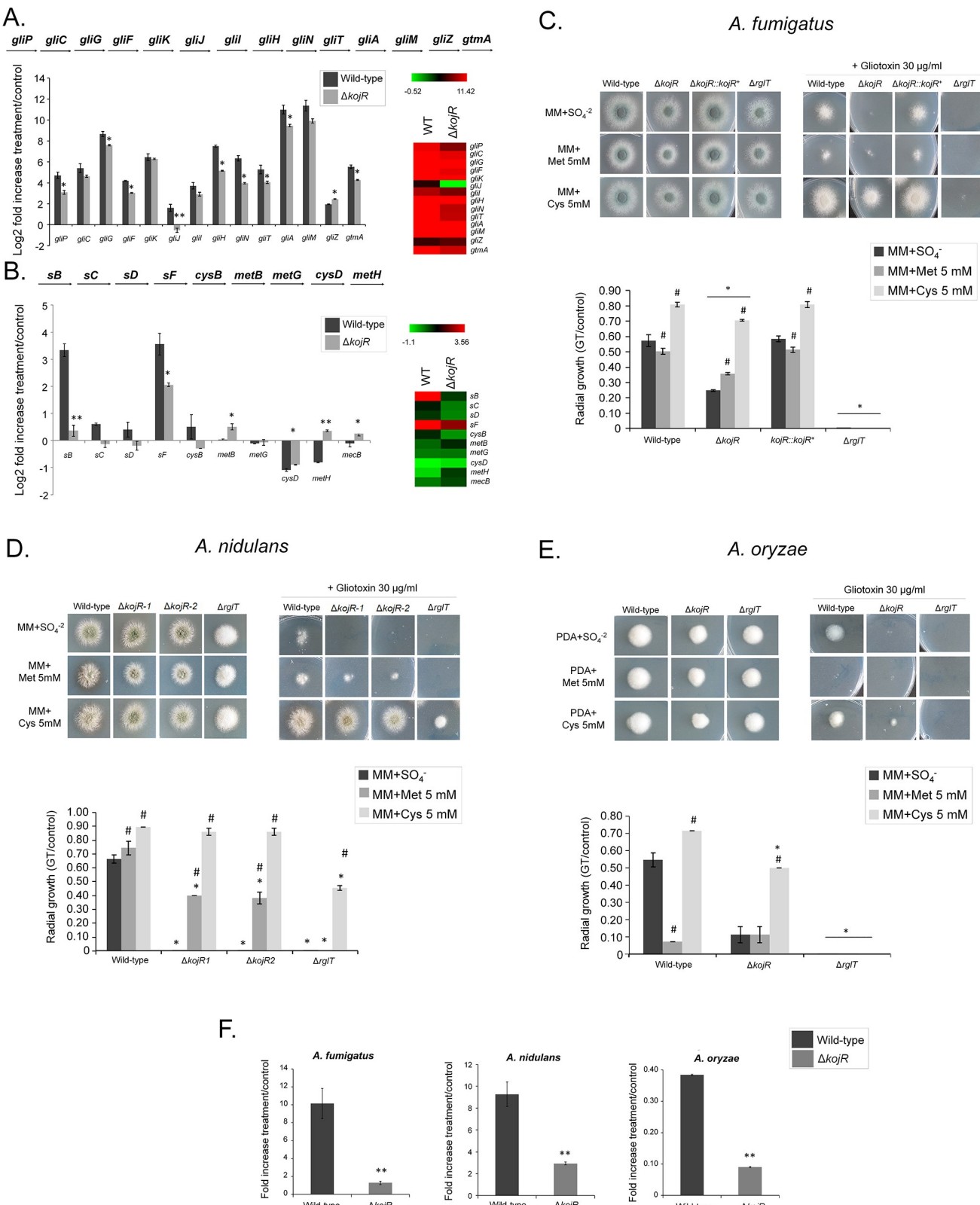

**Fig 11. *A. fumigatus gliJ* and sulfur assimilation genes are dowregulated in the Δ*kojR* mutant. – *A. fumigatus* was grown for** 21 h at 37°C before 5 μg/ml of GT was added to cultures for 3 hours. RNA was extracted from the wild-type and Δ*kojR* strains, reverse transcribed to cDNA and RT-qPCRs were peformed. Results are expressed as log2 fold increase and are the average of three biological replicates ± standard deviation. Statistical analysis was

performed using a one-tailed, paired t-test when compared to the control condition (**, $p < 0.01$ and *, $p < 0.005$). (A) Genes in the *A. fumigatus* GT pathway and *gtmA*. (B) Selected genes in the *A. fumigatus* sulfur assimilation and transsulfuration pathways. Heat map scale and gene identities are indicated. (C) *A. fumigatus*, (D) *A. nidulans*, and (E) *A. oryzae* wild-type, Δ*kojR*, Δ*kojR*::*kojR*+, and Δ*rglT* mutants were grown for 48 hours at 37°C on MM+SO$_4$$^{-2}$, MM+Met 5 mM, and MM+Cys 5 mM in the presence or absence of 10 or 30 μg/ml of GT. The results are the average of three repetitions ± standard deviation. Statistical analysis was performed using a one-tailed, paired t-test when compared to the control condition (*, $p < 0.005$ when comparing the mutant strains versus the wild-type and and #, $p < 0.005$ when comparing the growth on MM+Met or MM+Cys versus the growth on MM+SO$_4$$^-$). (F) **A. fumigatus, A. nidulans, and A. oryzae were grown for** 20 h at 37°C before 5 μg/ml of GT was added to cultures for 3 hours. RNA was extracted from the wild-type and Δ*kojR* strains, reverse transcribed to cDNA and RT-qPCRs were peformed. Results are the average of three biological replicates ± standard deviation. Statistical analysis was performed using a one-tailed, paired t-test when compared to the control condition (**, $p < 0.01$ and *, $p < 0.005$).

be suppressed by either methionine or cysteine as a single sulfur souce while *A. nidulans* Δ*rglT* GT susceptibility was partially suppressed by cysteine (Fig 11A–11E).

Taken together, these data indicate that although there is a global reduction in the expression of several genes in the GT BGC, including *gliT* (and also *gtmA*, not present in the GT BGC), *A. fumigatus* KojR is essential for the *gliJ* expression. Furthermore, *A. fumigatus* KojR is also important for the transcriptional modulation of several genes involved in inorganic sulfur assimilation and transsulfuration in the presence of GT. Cysteine suppression of GT susceptibility in *A. nidulans* and *A. oryzae* and trancriptional control of the *sA* sulfate permease in the presence of GT also indicated that KojR is involved in the regulation of the sulfur pathways in these species upon GT stress.

We also investigated if KojR could impact *A. fumigatus* virulence in a chemotherapeutic murine model of IPA. All mice infected with the WT and Δ*kojR*::*kojR* strains died between day 5 and day 7 post-infection (p.i.), whereas 10% of mice infected with the Δ*kojR* strain survived for the duration of the experiment (Figs 12A and S5). Fungal burden in the lungs was also significantly reduced for the Δ*kojR* strain after 3 days p.i. when compared to the WT and Δ*kojR*:: *kojR* strains (Fig 12B). In addition, histopathology of the lung tissue after 3 days p.i. with the Δ*kojR* strain demonstrated a significantly reduced inflammation score (Fig 12C) and immune cell recruitment to the tissue (Fig 12D) relative to the WT and Δ*kojR*::*kojR* strains. These results strongly indicate that KojR is important for *A. fumigatus* inflammation and virulence.

## Discussion

The transcriptional regulation of GT biosynthesis requires many components including TFs, protein kinases, G-protein signalling and chromatin modifying enzymes, which together integrate different signaling pathways [30]. In addition, GT biosynthesis requires GSH, of which the sulfur-containing amino acids methionine and cysteine are biosynthetic precursors; and GT biosynthesis is linked to cellular oxidative stress in mammalian and fungal cells via a yet to be described mechanism [31–35]. In addition, GT self-protection is essential during GT biosynthesis. The three main mechanisms of GT self-protection in *A. fumigatus* are the GT-neutralising oxidoreductase GliT, the GT efflux pump GliA, and the negative regulation of GT biosynthesis through GtmA [30]. RglT was recently identified as the main TF controlling the expression of *gliT* and of *gtmA* in *A. fumigatus*. The RglT homolog is also important for *gliT* expression and GT protection in the non-GT producer *A. nidulans* [42]. Despite this similarity, *A. nidulans* does not encode homologs of GtmA and GliA, suggesting that (i) these GT protection mechanisms are not required or that (ii) additional, unknown GT protection mechanisms exist in this fungus. It is perhaps not surprising that GtmA and GliA are not present in *A. nidulans*, because as a non-GT producing fungus, GtmA-mediated attenuation of GT biosynthesis and GT efflux are not required. We currently do not know how the presence of GT is signalled and/or whether this mycotoxin can be taken up by *A. nidulans*. In this work, the transcriptional response of *A. fumigatus* and *A. nidulans* upon prolonged exposure to GT was

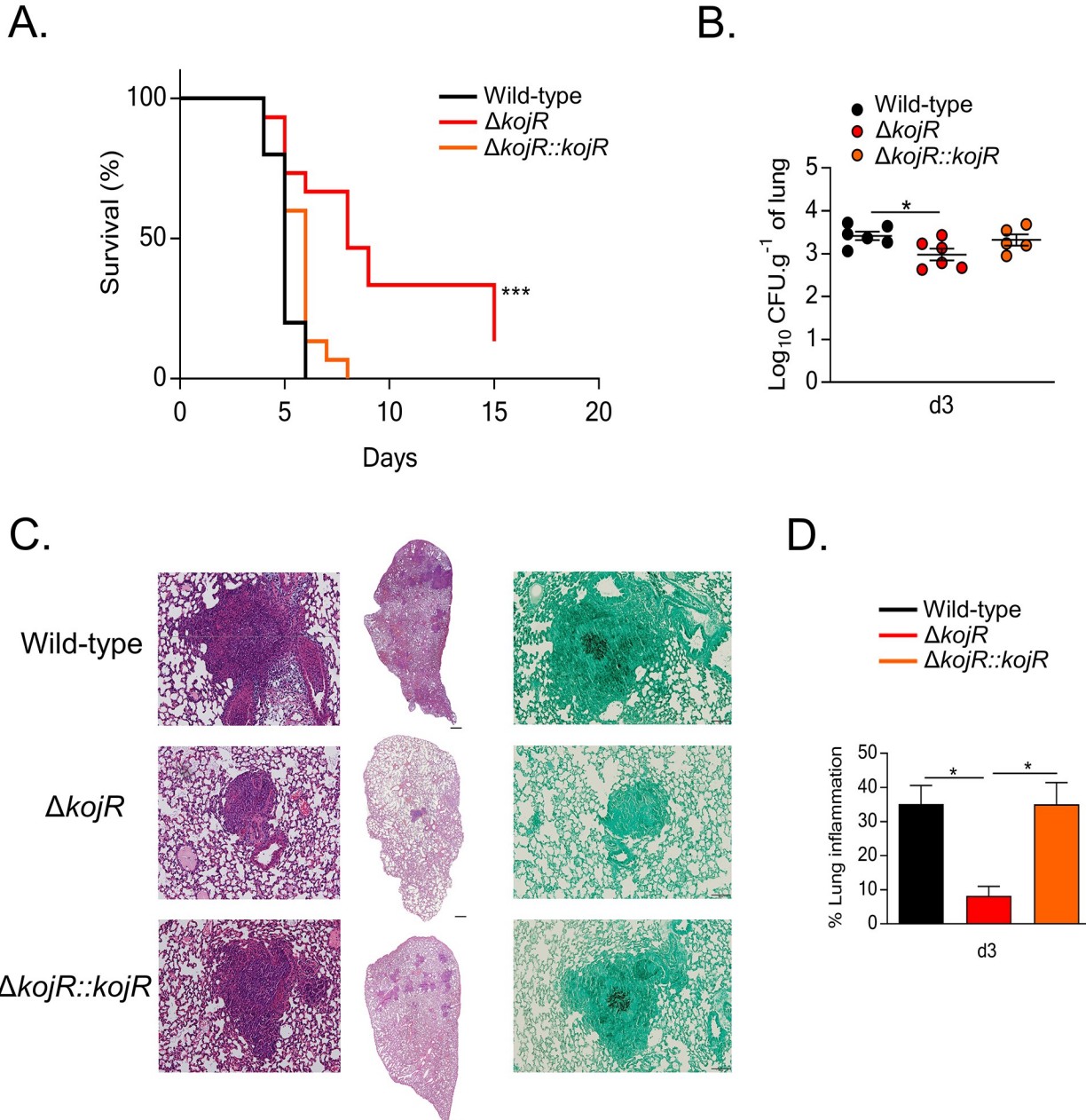

**Fig 12. KojR is required for virulence in a mouse model of invasive aspergillosis.** - **(A)** Survival curve (n = 10 mice/strain) infected with the indicated *A. fumigatus* strains. Phosphate buffered saline (PBS) was administered in a negative control group (n = 5). The indicated *P* values are based on the log-rank, Mantel-Cox, and Gehan-Breslow-Wilcoxon tests comparing the *KojR* deletion strain to the *WT* and complemented strains. **(B)** Fungal burden in murine lungs after 3 days post-infection (p.i.) with the different strains. Murine lungs were excised, ruptured, and resuspended before dilutions were prepared and incubated on plates containing complete medium. Fungal growth was assessed by counting the colony-forming units (CFU) on the plates for each dilution. **(C)** Histopathology of mice infected with the different strains. Lungs were excised at 3 days post-infection (p.i.) before lung sections were prepared and stained with hematoxylin and eosin (H&E) and/or with Gomori's methenamine silver (GMS). **(D)** Inflammation (expressed as %) in murine lungs after 3 days post-infection (p.i.) with the different strains. Murine lungs were excised, and slides of lungs sections were prepared. To quantify lung inflammation of infected animals, inflamed areas on slide images were analyzed using the thresholding tool in ImageJ software. Values are averages ± standard deviations (error bars) of three biological replicates (lungs from different mice). Values that are significantly different in a two-way multiple comparison ANOVA test are indicated as follows: $^*p$-value $< 0.05$, $^{***}p$-value $< 0.001$.

analysed and found to be significantly different between both fungal species. In agreement, the RglT-dependent regulome in both fungal species is also very different. This may be due to the evolutionary distance between both fungal species which share around 66–67% of amino acid identity, a protein identity comparable to the phylogenetic relationship between mammals and fish [53]. Despite this, our dataset also identified a conserved transcriptional response. These genes encode proteins important for gliotoxin modification, such as the observed RglT-dependent putative methyltransferase MtrA (AN3717/AFUA_6G12780) and one putative oxidoreductase (AN9051/AFUA_7G00700), attenuating directly or indirectly its activity. We have investigated one of these genes, *mtrA*, and demonstrated that it is important for GT protection in *A. fumigatus* and *A. nidulans*. The *A. nidulans* Δ*mtrA* is more sensitive to GT than the *A. fumigatus* Δ*mtrA* most likely because there are additional mechanisms of GT protection in *A. fumigatus*, such as GtmA. It remains to be investigated if MtrA is directly methylating GT or other proteins important for GT protection.

To further identify conserved components important for GT self-protection, we screened the *A. fumigatus* TF deletion library to identify TFs important for this process. We identified 15 TFs (4 have been characterized and 11 have not been characterized), whose deletion caused a significant decrease in growth in the presence of exogenous GT. The 11 uncharacterized TFs can broadly be classed into 4 categories based on their respective GT and bmGT biosynthesis profiles: (i) AFUA_2G17860 and AFUA_5G12060 (whose deletion mutants are GT non-producers and bmGT producers), (ii) AFUA_3G01100, AFUA_5G14390, AFUA_4G13060, AFUA_6G09870, AFUA_8G07280 and AFUA_8G05460 (whose deletion mutants are GT producers and bmGT overproducers), (iii) AFUA_8G03970 and *oefC* (AFUA_3G09670, whose deletion mutants are GT and bmGT overproducers) and (iv) AFUA_8G07360 (whose deletion mutant is a GT non-producer and bmGT overproducer). They present interesting targets for future mycotoxin-related studies.

In addition, our work identified an additional 4 TFs, which have previously been characterized, as important for GT biosynthesis. Yap1 (AFUA_6G09930, whose deletion mutant is a GT overproducer and bmGT overproducer) is an important regulator of resistance to oxidative stress [54]. SreA (AFUA_5G11260, whose deletion mutant is a GT non-producer and bmGT producer) regulates iron uptake, through repressing iron-regulated genes and siderophore biosynthesis genes in high-iron conditions [55]. The identification of these TFs emphasizes the close connection between GT production and oxidative stress and iron metabolism [30,34,48,49]. RgdA (AFUA_3G13920, whose deletion mutant is a GT non-producer and bmGT producer) has previously been reported as important for hyphal growth, asexual sporulation and virulence [56]. In contrast to our results, these authors observed increased GT production in the Δ*rgdA* mutant, which may be due to the fact they have used Af293 as a background strain for the *rgdA* deletion mutant whereas this work used CEA17 as a background strain [56]. Differences in cellular responses between these two *A. fumigatus* background strains have been reported before and include the role of the calcium-responsive TF CrzA during the caspofungin paradoxical effect [57]. KojR (AFUA_5G06800, whose deletion mutant is a GT non-producer and bmGT non-producer) is the *A. oryzae* homolog involved in the production of kojic acid [58]. It is perhaps not surprising that our work identified Yap1, SreA and RgdA as involved in GT biosynthesis and self-protection, as oxidative stress, iron metabolism, and fungal growth have all been shown to be important for GT and/or SM biosynthesis [30,34,48,49]. In contrast, the deletion of *kojR* was the only mutant strain where reduced amounts of GT and bmGT was observed, suggesting that this TF is important for GT metabolism. The role of this TF in GT protection was therefore further characterised in GT-producing *A. fumigatus* and non-producing (*A. nidulans* and *A. oryzae*) *Aspergillus* species.

These results expand significantly the number of TFs involved, and suggest additional complex mechanisms in regulating GT protection and production.

As a preliminary step to understand additional KojR gene targets, we performed extensive RTqPCR experiments with all the *gli* BGC genes, *gtmA*, and selected genes involved in the inorganic sulfur assimilation and transsulfurattion pathways. Our results showed a global reduction of 20 to 30% of the expression of the *gli* genes when the Δ*kojR* was exposed to GT, except for *gliJ*, encoding a membrane dipeptidase, that is significantly down-regulated in the mutant. These results strongly indicate that *gliJ* is a direct or indirect target for *A. fumigatus* KojR. The blockage of the GT biosynthesis pathway caused by the lack of *gliJ* expression helps to understand the lack of GT production in this mutant. As previously shown [45], we have also observed that exposure of *A. fumigatus* to GT increases the expression of genes involved in the inorganic sulfur assimilation and transsulfuration pathways. KojR is also negatively affecting the increased mRNA accumulation of several genes in these pathways upon GT exposure. This could impact the S-adenosyl methione (SAM):S-adenosyl homocysteine (SAH) balance, essential for the methylation of GT by GtmA, helping to explain why there is a decrease in the bmGT production in the *A. fumigatus* Δ*kojR* mutant. Interestingly, cysteine supplementation (instead of inorganic sulfur) can suppress the Δ*kojR* GT-sensitivity, emphasizing the role played by KojR in sulfur assimilation. The main TF for sulfur inorganic assimilation, MetR, is transcriptionally induced in the presence of GT [33] and Δ*metR* showed significantly increased sensitivity to GT [59]. It remains to be investigated a possible interaction between KojR and MetR since we observed decreased mRNA accumulation of *metR* upon GT in the Δ*kojR* mutant.

We have also shown that KojR is essential for virulence in a chemotherapeutic murine model of IPA. Both Δ*kojR* and Δ*gliP* (*gliP* encodes the nonribosomal peptide synthase of GT BGC) do not produce GT but in contrast to Δ*kojR*, Δ*gliP* is still virulent in the chemotherapeutic murine model of IPA [60]. Previously, we observed that Δ*rglT* is hypovirulent in the same murine model and suggest that these virulence defects could be due to defects in oxidative stress resistance [42]. It is possible the impact of the lack of KojR in the sulfur metabolism affects the growth and virulence of Δ*kojR* in the murine lungs. Additional experiments are necessary to clarify this and it remains subject to future investigations.

We were able to show a conserved mechanism of protection from GT in these three fungi whereby deletion of *kojR* and *rglT* increases sensitivity to exogenous GT. This protection is likely to occur through the RglT-dependent regulation of GliT, where expression of *gliT* is dependent on RglT in the presence of GT in the three fungal species. Future studies will determine growth of the respective *gliT* deletion strains in order to confirm this conserved mechanism of GT protection in *Aspergillus* species. The expression of *gliT* is also dependent on KojR in all three fungi in the presence of GT; however, KojR is partially required for *gliT* expression in *A. fumigatus* and *A. oryzae*, whereas KojR represses *gliT* in *A. nidulans*, which does not produce GT. These results suggest that KojR-dependent regulation of *gliT* differs between GT-producing and non-producing *Aspergillus* species and these results are in agreement with our *A. fumigatus*/*A. nidulans* RNA-seq dataset. In *A. fumigatus* and *A. oryzae*, *rglT* expression is dependent on KojR and we suggest here that KojR directly regulates *rglT* and indirectly *gliT*. This regulatory hierarchy would therefore confer protection from GT in these fungi. In *A. fumigatus*, *kojR* expression is dependent on RglT whereas in *A. oryzae kojR* repression is dependent on RglT. These results suggest that differences in the regulation of both TFs in these fungi.

Interestingly, we observed that both *A. fumigatus* and *A. oryzae* Δ*rglT* and Δ*kojR* are more sensitive to GT and involved in the positive regulation of the oxidoreductase *gliT* and *gtmA*. Although *gliT* expression is reduced in *A. nidulans* Δ*rglT*, it is increased in Δ*kojR*, suggesting

the mechanism of *gliT* regulation is distinct in this species. Since GliT is essential for *A. nidulans* GT detoxification, these results indicate that increased expression of *gliT* in the *A. nidulans* Δ*kojR* strain is not enough to confer GT resistance to *A. nidulans*. These results suggest that KojR is also important for GT detoxification in a GT non-producer and that there are additional mechanisms of GT detoxification regulated by KojR in *A. nidulans*. Actually, two pieces of evidence suggest that KojR is also affecting the inorganic sulfur assimilation in *A. nidulans* and *A. oryzae*: (i) the *sB* sulfate permease gene is down-regulated in the presence of GT, and (ii) cysteine as a single sulfur source can suppress GT toxicity in these two species. We hypothesize that cysteine can suppress GT susceptibility in all three species because most likely it can bypass the initial steps of inorganic sulfur assimilation, providing directly cysteine for the synthesis of ergothioneine, glutathione, and methionine. Ergothioneine and glutathione are essential for oxiredox balance in the cell and also for GT prodution while methione can provide methyl groups for GT methylation, crucial for bmGT production. Interestingly, cysteine as a single sulfur source can suppress *A. nidulans* Δ*rglT* GT sensitivity but not *A. fumigatus* and *A. oryzae* Δ*rglT* GT sensitivity, suggesting diferences in the organic sulfur assimilation in these species during GT detoxification.

Although *A. oryzae* has the GT BGC, we were unable to detect GT production in the *A. oryzae* wild-type, Δ*rglT* and Δ*kojR* strains (S4 Fig). However, RglT and KojR are still regulating *A. oryzae gliT* and *gtmA*. *A. oryzae* strains have been selected for millenia for the saccharification of starch-rich rice to sugars that are fermentable by *Saccharomyces cerevisiae* into sake, the high-alcohol rice wine [61]. The *A. oryzae*-rice mixture (*koji*-culture) is mixed with additional steamed rice and fermented by *S. cerevisiae*. This *A. oryzae* rice starch breakdown and subsequent conversion into alcohol by *S. cerevisiae* occurs side by side [61]. Extensive genome analysis suggested that, through selection by humans, atoxigenic lineages of *A. oryzae* evolved into "cell factories" important for the saccharification process [62]. Interestingly, production of the secondary metabolite kojic acid was retained in these strains during the *koji*-culture [58]. The kojic acid gene cluster is one of the simplest clusters of secondary metabolite production ever reported, composed by three genes: an enzyme containing a FAD-dependent oxidoreductase domain, a transporter, and a *kojR* encoding a GAL4-like $Zn(II)_2Cys_6$ transcription factor [58,63]. As expected, *A. oryzae* Δ*kojR* is unable to produce kojic acid [58] but there is no involvement of RglT in kojic acid production.

There is extensive heterogeneity in drug susceptibility, nutritional requirements, and virulence across *A. fumigatus* clinical and environmental isolates [57,64–66]. It remains to be determined if GT-sensitive mutants in different *Aspergillus spp* isolates will behave accordingly to what was observed in this manuscript using the available isolates. Our work provides new opportunities to understand GT production and protection, and also highlights significant differences in transcriptional responses to extracellular mycotoxins between fungal species. More importantly, the identification of a conserved transcriptional response to exogenous GT in different *Aspergillus* species and isolates provides a basis for future studies which will further decipher these protection pathways.

## Materials and methods

### Ethics statement

The animal procedures were designed as a prospective, randomized, blinded, experimental study, and performed following the guidelines for animal research and the 3R principles of the EU directive. Eight-week-old gender and age-matched C57Bl/6 mice were bred under specific-pathogen-free conditions and kept at the Life and Health Sciences Research Institute (ICVS) Animal Facility. Animal experimentation was performed following biosafety level 2 (BSL-2),

and protocols were approved by the Institutional Animal Care and Use Committee (IACUC) of the University of Minho. The ethical and regulatory approvals were consented by Ethics Committee for Research in Life and Health Sciences of the University of Minho (SECVS 074/2016). Furthermore, all *in vivo* procedures followed the EU-adopted regulations (Directive 2010/63/EU) and were conducted according to the guidelines sanctioned by the Portuguese ethics committee for animal experimentation, Direção-Geral de Alimentação e Veterinária (DGAV).

## Strains and media

All strains used in this study are listed in S6 Table. Strains were grown at 37°C except for growth on allyl alcohol-containing solid medium, which was carried out at 30°C. Conidia of *A. fumigatus* and *A. nidulans* were grown on complete medium (YG) [2% (w/v) glucose, 0.5% (w/v) yeast extract, trace elements] or minimal media (MM) [1% (w/v) glucose, nitrate salts, trace elements, pH 6.5]. Solid YG and MM were the same as described above with the addition of 2% (w/v) agar. Where necessary, uridine and uracil (1.2 g/L) were added. Trace elements, vitamins, and nitrate salt compositions were as described previously [67]. Potato Dextrose Agar (PDA, Difco) was used for *A. oryzae*. For iron or zinc starvation or excess experiments, strains were growth in solid MM without $FeSO_4$ or zinc. For phenotypic characterization, plates were inoculated with $10^4$ spores per strain and left to grow for 120 h at 37 or 30°C. All radial growth experiments were expressed as ratios, dividing colony radial diameter of growth in the stress condition by colony radial diameter in the control (no stress) condition.

## Phylogenomic inference of Aspergillus–Penicillium species phylogeny

Publicly available gene annotations for *Aspergillus* and *Penicillium* species (n = 55) were downloaded from NCBI in February 2021. OrthoFinder, v2.3.8 [68], was used to identify orthologous groups of genes using the protein sequences of the 55 proteomes as input. From the 18,979 orthologous groups of genes, 1,133 were identified to be single-copy and present in all proteomes. All 1,133 genes were aligned using MAFFT, v7.402 [69], with the 'auto' option. The corresponding nucleotide sequences were threaded onto the protein alignments using the 'thread_dna' function in PhyKIT, v1.1.2 [20]. The resulting nucleotide alignments were trimmed using ClipKIT, v1.1.3 [26], with the 'smart-gap' option. All 1,133 genes were concatenated into a single matrix with 2,446,740 sites using the 'create_concat' function in PhyKIT [20]. The resulting alignment was used to reconstruct the evolutionary history of the 55 species using IQ-TREE, v2.0.6 [70]. During tree search, the number of trees maintained during maximum likelihood inference was increased from five to ten using the 'nbest' option. Bipartition support was assessed using 1,000 ultrafast bootstrap approximations [71].

Comparison of the inferred phylogeny showed that it was nearly identical to the phylogeny inferred in a previous genome-scale examination of evolutionary relationships among *Aspergillus* and *Penicillium* species [44]. Minor differences were observed among bipartitions previously determined to harbor high levels of incongruence. These differences concern the placements of *Penicillium decumbens* and *Aspergillus awamori*. Differences in gene and taxon sampling may be responsible for these incongruences. All bipartitions received full support.

## Identifying homologous biosynthetic gene clusters and orthologs of *RglT* and *kojR*

To identify homologs of the biosynthetic gene clusters involved in the production of gliotoxin and kojic acid, the 13 protein sequences of the gliotoxin cluster from *Aspergillus fumigatus* and the three protein sequences of the kojic acid cluster from *Aspergillus oryzae* were used as

queries during sequence similarity searches to identify homologs in the 55 proteomes. Sequence similarity searches were conducted using NCBI's blastp function from BLAST+, v 2.3.0 [72], with an expectation value threshold of 1e-4. Across all homologs, the physical proximity of each gene was assessed. We considered biosynthetic gene clusters to be homologous if 7 / 13 genes from the gliotoxin biosynthetic gene cluster were present including the nonribosomal peptide synthase, *gliP*. For the kojic acid biosynthetic gene cluster, we required that 2 / 3 genes from the kojic acid cluster were present including *kojA*, the oxidoreductase required for kojic acid production [73]. When examining gene proximity, a gene boundary distance of five genes was used. To identify genes orthologous to *RglT* and *kojR*, OrthoFinder results were parsed to identify genes in the same orthologous group as each gene.

## Testing the association of the phylogenetic distributions of *kojR* and *RglT*

To evaluate if the distributions of *kojR*, *RglT*, and their BGCs on the species phylogeny are significantly associated, we examined the correlation of the phylogenetic distribution of the presence and absence patterns of the two genes on the species phylogeny. We used Pagel's binary character correlation test implemented in phytools, v0.7–70 [74] to examine four scenarios: the distributions of two genes / BGCs are not correlated (scenario A), the distribution of one gene is correlated with the distribution of another gene / BGC but not vice versa (scenarios B and C), and the distributions of two genes / BGCs are correlated (scenario D). The best fitting model was then determined using the weighted Akaike information criterion (AIC).

## Mouse model of pulmonary aspergillosis

The *A. fumigatus* Δ*ku80 pyrG*+ CEA17 and Δ*kojR* strains were grown on malt extract agar for 7 days before infection. To induce immunosuppression, 150 mg/Kg of cyclophosphamide (Sigma) was administrated intraperitoneally on days -4, -1, and +2, while 200 mg/Kg of cortisone 21-acetate (Acros Organics) was administrated on day -3 via the subcutaneous route. On day 0, mice were challenged intranasally with $1\times10^5$ live conidia following anesthesia with 75 mg/Kg of ketamine (Ketamidor, Richter Pharma) and 1 mg/Kg of medetomidine (Domtor, Ecuphar). To avoid bacterial infections, animals were treated with 50 µg/mL of chloramphenicol in drinking water ad libitum. For the survival studies, animals (n = 10) were monitored twice daily for 15 days after infection. Animals were weighted and sacrificed in case of 20% loss weight, severe ataxia or hypothermia, and other severe complications. In some experiments, mice (n = 5) were euthanized on day +3 and their lungs were aseptically removed, weighed, and homogenized in 2 mL sterile PBS in a tissue homogenizer (Ultra-Turrax T25 Basic, IKA Works, Inc.). For fungal burden assessment, lung homogenates were plated on Sabouraud 4% dextrose agar (Sigma) supplemented with chloramphenicol at 50 µg/mL (Sigma). For histological analysis, lungs were excised and fixed with 10% buffered formalin solution (Sigma) for at least 48 hours, processed and paraffin embedded. Lung paraffin sections were stained with Hematoxylin and Eosin (H&E) or Gomori's Methenamine silver (GMS) (Sigma) for pathological examination. Images were acquired using a widefield upright microscope BX61 (Olympus) and a DP70 high-resolution camera (Olympus). Morphometry analyses were carried out on slide images using ImageJ software (v1.50i, NIH, USA).

## Generation of deletion mutants for A. *oryzae kojR* and *rglT*, A. *nidulans kojR* and *mtrA*, and A. *fumigatus* Δ*kojR::kojR*+ and *mtrA*

The DNA fragment for *kojR* deletion, using the *A. oryzae* orotidine-5′-decarboxylase gene (*pyrG*) as a selectable marker, was generated by fusion polymerase chain reaction (PCR) as follows: DNA fragments upstream and downstream of the *kojR* coding region were amplified by

PCR using *A. oryzae* genomic DNA as a template and the primer sets BR-TF040-L-F + BR-TF040-L-R and BR-TF040-R-F + BR-TF040-R-R, respectively. The *pyrG* fragment was amplified by PCR using *A. oryzae* genomic DNA as a template and the primers BR-TF040-P-F and BR-TF040-P-R. Then, these three PCR fragments were mixed, and a second round of PCR with the primers BR-TF040-L-F and BR-TF040-R-R was performed. For *rglT* deletion, DNA fragments upstream and downstream of the *rglT* coding region were amplified by PCR using the primer sets BR-TF390-L-F + BR-TF390-L-R and BR-TF390-R-F + BR-TF390-R-R, respectively. The *pyrG* fragment was amplified by PCR using the primers BR-TF390-P-F and BR-TF390-P-R. These three PCR fragments were mixed, and a second round of PCR without primers was carried out. Then, a third round of PCR with BR-TF390-L-F_Nest and BR-TF390-R-R_Nest was performed. The resultant PCR fragments were introduced into the recipient strain RkuN16ptr1 (*Δku70::ptrA*, *pyrG⁻*) according to [75].

In *A. nidulans*, the *kojR* and *mtrA* genes were deleted using a gene replacement cassettes which were constructed by *"in vivo"* recombination in *S. cerevisiae* as previously described by [76]. Thus, approximately 1.0 kb from each 5'-UTR and 3'-UTR flanking region of the targeted ORFs regions were selected for primer design. The primers AN4118 (kojR)/mtrA_pR-S426_5UTR_fw and AN4118/mtrA_pRS426_3UTR_rv contained a short homologous sequence to the MCS of the plasmid pRS426. Both the 5- and 3-UTR fragments were PCR-amplified from *A. nidulans* genomic DNA (TNO2A3 strain). The *pyrG* gene placed within the cassette as a prototrophic marker was amplified from pCDA21 plasmid using the primers AN4118/mtrA_5UTR_pyrG_rv and AN4118/mtrA_3UTR_pyrG_fw. The cassette was PCR-amplified from these plasmids utilizing TaKaRa Ex Taq DNA Polymerase (Clontech Takara Bio) and used for *A. nidulans* transformation.

Complementation of the kojR gene in *A. fumigatus* was achieved via cotransformation. A fragment containing the kojR gene flanked by 1-kb 5′ and 3′ UTR regions was PCR amplified (primers AfukojR complemented 5UTR pRS fw and AfukojR complemented 3UTR pRS rv). This DNA fragment was transformed into the Δ*kojR* strain along with the plasmid pPTR1 (TaKaRa Bio) containing the pyrithiamine resistance ptrA gene. Mutants were selected on MM supplemented with 1 μg ml−1 pyrithiamine and confirmed with PCR using the external primers (AfukojR 5UTR ext fw and AfukojR 3UTR ext rv). Primer sequences are listed in S7 Table. *A. fumigatus mrtA* gene was deleted using a gene replacement cassette which was also constructed by *"in vivo"* recombination in *S. cerevisiae*. Approximately 1.0 kb from each 5'-UTR and 3'-UTR flanking region of the targeted ORF regions were selected for primer design. The primers Afu6g12780_pRS426_5UTR FW and Afu6g12780_pRS426_3UTR RV contained a short homologous sequence to the MCS of the plasmid pRS426. The *pyrG* gene placed within the cassette as a prototrophic marker was amplified from pCDA21 plasmid using the primers Afu6g12780_5UTR_pyrG_RV and Afu6g12780_3UTR_pyrG_FW. Southern blot analysis to confirm the *mtrA* deletion (S5 Fig). Primer sequences are listed in S7 Table.

## Screening of A. fumigatus transcription factor deletion strains

A library of 484 *A. fumigatus* TFKO (transcription factor knock out) strains [46] was screened for sensitivity to gliotoxin. To verify this sensitivity, gliotoxin concentrations lower than the minimum inhibitory concentration (MIC) for the *A. fumigatus* wild type strain, CEA17, (70μg / mL) [42] were used. Conidia were inoculated in 96-well plates at the final concentration of $10^4$ spores / well, in 200 μl MM supplemented or not with concentrations of gliotoxin ranging from 35 μg/ml (0.5 x MIC of wt) to 70 μg/ml, and incubated at 37˚C for 48 h. Three independent experiments were carried out. Validation of this screening was carried out by growing sensitive strains at the concentration of $10^4$ spores in solid MM plates with 30 μg/ml gliotoxin at 37˚C, for 48 h.

## Gliotoxin and Bis(methylthio)gliotoxin Presence Analysis by ultra-high-performance liquid chromatography (UHPLC) coupled to high-resolution electrospray ionization mass spectrometry (HRESIMS)

All strains were grown on Czapek Dox Agar (CDA) at 37˚C in an incubator (VWR International) in the dark over four days. To evaluate the biosynthesis of gliotoxin (GT) and bis(methylthio)gliotoxin (bmGT) in these fungal strains, cultures were extracted with organic solvents and analyzed by mass spectrometry (see below). The agar plates were sprayed with MeOH, chopped with a spatula, and then the contents were transferred into a scintillation vial. Afterwards, acetone (~15 ml) was added to the scintillation vial, and the resulting slurry was vortexed vigorously for approximately 3 min and then let too set at RT for 4 h. Next, the mixture was filtered, and the resulting extract was dried under a stream of nitrogen gas.

HRESIMS experiments utilized a Thermo QExactive Plus mass spectrometer (Thermo Fisher Scientific), equipped with an electrospray ionization source. This was coupled to an Acquity UHPLC system (Waters Corp.), using a flow rate of 0.3 ml/min and a BEH $C_{18}$ column (2.1 mm x 50 mm, 1.7 µm) that was operated at 40˚C. The mobile phase consisted of $CH_3CN$–$H_2O$ (Fischer Optima LC-MS grade; both acidified with 0.1% formic acid). The gradient started at 15% $CH_3CN$ and increased linearly to 100% $CH_3CN$ over 8 min, where it was held for 1.5 min before returning to starting conditions to re-equilibrate.

Extracts were analyzed in biological triplicates and each of these in technical triplicate using selected ion monitoring (SIM) in the positive ion mode with a resolving power of 35,000. For GT, the SIM had an inclusion list containing the mass $[M+H]^+$ = 327.04677, with an isolation window of 1.8 Da and a retention window of 2.5–5.0 min. For bmGT, the SIM had an inclusion list containing the mass $[M+H]^+$ = 357.09373, with an isolation window of 1.8 Da and a retention window of 2.5–4.5 min. The extracts, gliotoxin standard (Cayman Chemical), and bis(methylthio)gliotoxin standard (isolated previously) [26] were each prepared at a concentration of 2.5 mg/ml for the extracts and 0.01 mg/ml for the standards; all extracts were dissolved in MeOH and injected with a volume of 3 µl. To eliminate the possibility for sample carryover, two blanks (MeOH) were injected between every sample, and the standards were analyzed at the end of of the run.

To ascertain the relative abundance of GT and bmGT in these samples, batch process methods were run using Thermo Xcalibur (Thermo Fisher Scientific). For GT, we used a mass range of 327.0437–327.0492 Da at a retention time of 3.24 min with a 5.00 second window. For bmGT, we used a mass range of 357.0928–357.9937 Da at a retention time of 3.30 min with a 5.00 second window.

## RNA extraction, RNA-sequencing, cDNA synthesis and RT-qPCR

All experiments were carried out in biological triplicates and conidia ($10^7$) were inoculated in liquid MM (*A. fumigatus* and *A. nidulans*), or in Potato Dextrose Broth (*A. oryzae*). Gliotoxin was added for 3 h to the culture medium to a final concentration of 5 µg/ml after strains were grown for 21 h in MM. As GT was dissolved in DMSO, control cultures received the same volume of DMSO for 3 h. For total RNA isolation, mycelia were ground in liquid nitrogen and total RNA was extracted using TRIzol (Invitrogen), treated with RQ1 RNase-free DNase I (Promega), and purified using the RNAeasy kit (Qiagen) according to the manufacturer's instructions. RNA was quantified using a NanoDrop and Qubit fluorometer, and analyzed using an Agilent 2100 Bioanalyzer system to assess the integrity of the RNA. All RNA had an RNA integrity number (RIN) between 8.0 and 10 (Thermo Scientific) according to the manufacturer's protocol.

RNA-sequencing was carried out using Illumina sequencing. The cDNA libraries were constructed using the TruSeq Total RNA with Ribo Zero (Illumina, San Diego, CA, USA). From

0.1–1 μg of total RNA, the ribosomal RNA was depleted and the remaining RNA was purified, fragmented and prepared for complementary DNA (cDNA) synthesis, according to manufacturer recommendations. The libraries were validated following the Library Quantitative PCR (qPCR) Quantification Guide (Illumina). Paired-end sequencing was performed on the Illumina NextSeq 500 Sequencing System using NextSeq High Output (2 x 150) kit, according to manufacturer recommendations. The BioProject ID in the *NCBI's BioProject database* is PRJNA729661.

For RT-qPCR, the RNA was reverse transcribed to cDNA using the ImProm-II reverse transcription system (Promega) according to manufacturer's instructions, and the synthesized cDNA was used for real-time analysis using the SYBR green PCR master mix kit (Applied Biosystems) in the ABI 7500 Fast real-time PCR system (Applied Biosystems, Foster City, CA, USA). Primer sequences are listed in S7 Table. RNASeq data was processed to select genes for normalization of the qPCR data, applying the method of [77]. Briefly, genes with a coefficient of variation smaller than 10%, behaving in a normally distributed manner, and with large expression values (FPKM) were selected. After experimental validation, Afu2g02680 (Putative matrix AAA protease) and AN1191 [Small ubiquitin-like modifier (SUMO) protein]. For *A. oryzae* experiments, *tubA* (AO090009000281) was used as a normalizer.

## Kojic Acid (KA) production by A. *oryzae*

KA production by the A. oryzae strains was detected by the colorimetric method verifying the intensity of the red color formed by chelating ferric ions [58,78]. The standard medium for KA production [0.25% (w/v) yeast extract, 0.1% (w/v) $K_2HPO_4$, 0.05% (w/v) $MgSO4-7H_2O$, and 10% (w/v) glucose, pH 6.0] containing 1 mM ferric ion ($FeCl_3$) was used for detecting KA production by *A. oryzae*. The color of the medium turned red, indicating the presence of KA chelated with ferric ions. This color change could be observed at around 5 days after cultivation.

## Supporting information

**S1 Fig. Comparison of RT-qPCR-based expression profiles of selected *A. fumigatus* and *A. nidulans* genes that were significantly modulated in the RNA-sequencing experiments.** (PDF)

**S2 Fig. Growth of the wild-type and GT-sensitive mutants in MM with different sulfur sources. The strains were grown for 72 hs at 37˚C.** (PPTX)

**S3 Fig. Southern blot analysis to confirm the *mtrA* deletion.** (PDF)

**S4 Fig. Base peak chromatograms of the strain Af293, the *A. oryzae* Δ*kojR* strain, the *A. oryzae* Δ*rglT* strain, and the wild-type (WT) *A. oryzae* strain grown on CDA (Czapek-Dox Agar, induces gliotoxin biosynthesis) at 37˚C.** Results show that the *A. oryzae* WT and deletion strains do not produce (A) gliotoxin or (B) bis(methylthio)gliotoxin. The data are presented as extracted ion chromatograms (XIC) from selected ion monitoring (SIM) using the protonated mass of gliotoxin ($C_{13}H_{15}N_2O_4S_2$; $[M+H]^+$ = 327.0473) and bis(methylthio)gliotoxin ($C_{15}H_{21}N_2O_4S_2$; $[M+H]^+$ = 357.0928) with a window of ± 5.0 ppm. GT and bmGT were detected in the *A. fumigatus* positive control strain Af293. (PDF)

**S5 Fig. Virulence of *A. fumigatus* Δ*kojR* mutant.** (PPTX)

**S1 Table. Genes significantly differentiallly expressed in the *A. fumigatus* wild-type strain exposed to GT in comparison to the GT-free, control condition.**
(XLS)

**S2 Table. Genes significantly differentiallly expressed in the *A. fumigatus* Δ*rglT* strain in comparison to the wild-type strain when exposed to GT.**
(XLSX)

**S3 Table. Genes significantly differentiallly expressed in the *A. nidulans* wild-type strain exposed to GT in comparison to the GT-free, control condition.** If applicable, the *A. fumigatus* homologue is also indicated.
(XLSX)

**S4 Table. Genes significantly differentiallly expressed in the *A. nidulans* Δ*rglT* strain in comparison to the wild-type strain when exposed to GT.** If applicable, the *A. fumigatus* homologue is also indicated. Also shown are a list of genes that are dependent or independent on RglT for their expression.
(XLSX)

**S5 Table. GT and bmGT production in the *A. fumigatus* wild-type and transcription factor deletion strains.**
(XLS)

**S6 Table. Strains used in this work.**
(DOCX)

**S7 Table. List of primers used in this work.**
(DOCX)

## Author Contributions

**Conceptualization:** Gustavo H. Goldman.

**Data curation:** Maria Augusta Crivelente Horta.

**Formal analysis:** Maria Augusta Crivelente Horta, Nicholas H. Oberlies, Jacob L. Steenwyk, Antonis Rokas, Laure N. A. Ries.

**Funding acquisition:** Nicholas H. Oberlies, Gustavo H. Goldman.

**Investigation:** Patrícia Alves de Castro, Ana Cristina Colabardini, Maísa Moraes, Sonja L. Knowles, Huzefa A. Raja, Relber A. Gonçales, Cláudio Duarte-Oliveira, Agostinho Carvalho, Laure N. A. Ries.

**Project administration:** Gustavo H. Goldman.

**Resources:** Yasuji Koyama, Masahiro Ogawa, Katsuya Gomi.

**Supervision:** Gustavo H. Goldman.

**Writing – original draft:** Laure N. A. Ries, Gustavo H. Goldman.

**Writing – review & editing:** Patrícia Alves de Castro, Ana Cristina Colabardini, Maísa Moraes, Maria Augusta Crivelente Horta, Sonja L. Knowles, Huzefa A. Raja, Nicholas H. Oberlies, Yasuji Koyama, Masahiro Ogawa, Katsuya Gomi, Jacob L. Steenwyk, Antonis Rokas, Relber A. Gonçales, Cláudio Duarte-Oliveira, Agostinho Carvalho, Laure N. A. Ries.

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
