## [Decision Letter · Decision Letter 0]

20 Dec 2021

Dear Dr Goldman,

Thank you very much for submitting your Research Article entitled 'Regulation of gliotoxin biosynthesis and protection in Aspergillus species' to PLOS Genetics.

The manuscript was fully evaluated at the editorial level and by independent peer reviewers. The reviewers appreciated the attention to an important topic but identified some concerns that we ask you address in a revised manuscript. We therefore ask you to modify the manuscript according to the review recommendations. Your revisions should address the specific points made by each reviewer.

1) Provide a list of your responses to the review comments and a description of the changes you have made in the manuscript.  The comments are generally quite minor so the responses need not be detailed.

[LINK]

Yours sincerely,

Aaron P. Mitchell, PhD

Associate Editor

PLOS Genetics

Gregory P. Copenhaver

Editor-in-Chief

PLOS Genetics

Reviewer's Responses to Questions

**Comments to the Authors:**

Reviewer #1: The article presented here, further provides within the mechanisms behind the regulation of gliotoxin in the human pathogenic fungus Aspergillus fumigatus. Furthermore, the article presents a new transcription factor involved in the defense during the exposure of this relevant mycotoxin. Gliotoxin toxicity is very unspecific, affecting protein folding, oxidative stress, methylation and probably chromatin structures and global gene expression. Therefore, it is not surprising that a large number of transcription factors are directly and indirectly influenced by the exogenous exposure of this toxin. However, the strength of the article is to include in the study also gliotoxin non-producing fungi, pointing to interspecies defense mechanisms.

Overall, the article was well presented, even if the amount of the data reported is massive. This obviously makes reading and understanding is not easy. Nevertheless, the efforts made by the authors are appreciable. Since the authors have already addressed an extensive revision process, I have no further experiments to ask, but I would appreciate some changes in the text.

Lines 74-75: The authors wrote: “KojR regulates the expression of another TF, an oxidoreductase, …”. This sentence is confusing, it seems like KojR is an oxidoreductase. Please change.

Line 158: Please remove “and is distantly related to A. fumigatus”; it is redundant with line 165.

Lines 172-175. This paragraph is confusing. I would suggest: We found that one of these TFs is a KojR hortolog, previously reported as regulator of the kojic acid production gene cluster in A. oryzae, is important as well for A. nidulans and A. oryzae GT protection and involved in A. fumigatus virulence, GT self-protection and GT and bmGT biosynthesis. Or something similar…

Lines 272-279: Is it necessary to list all those genes here? A potential reader can see them on the figure… The same for lines 327-334.

Lines 351-352: Why is it interesting? I could not get it.

Line 633: according to your results reported in figure 6 there is still a minor production of gliotoxin in the Delta-kojR mutant strain. Am I wrong?

Line 715-717: this statement is very speculative. If there are no data supporting that, please remove it.

Lines 739-741: how this work may lead to the development of anti-fungal strategies? Sounds like a "pret-a-porter" closing remark.

Reviewer #2: This revised manuscript is very nice. My comments are minor:

1) The sentence in lines 420-422 needs to be re-written because it seems to imply that deletion of kojR results in resistance to GT. For example, the sentence could be changed to: In contrast to deletion of rglT, which made all three species fully sensitive to GT, deletion of kojR resulted in only partial sensitivity to GT.

2)Fig. 8E is mentioned before Fig. 8D in the text. These two figures should be switched.

Reviewer #3: In this manuscript, de Castro and colleagues utilize transcriptomics and a knockout library screen to identify transcription factors with a previously unidentified role GT production and protection in Aspergillus. Overall, the data are thorough and well-presented, and the manuscript itself is well-written. While I do not have any experimental suggestions or concerns, I do have a few editorial notes that should be considreed to improve clarity or bolster the discussion. These are outlined below and in the attached document.

Major comments:

mtrA nomenclature: The ∆mtrA1 and ∆mtrA2 nomenclature (Figure 3) gives the impression that these are deletions in separate genes, i.e. paralogs of mtrA. To indicate independent mutants in the same gene, a better system would be (∆mtrA-1, ∆mtrA-2 or ∆mtrA #1, mtrA #2).

Figure 6: What exactly is being plotted in the heatmap (Figure 6c)? Is this a relative fold change between WT and the respective mutants? Some specific points of confusion for me include:

- 5G14390 and 6G09870 are both bright yellow for GT whereas WT is black. This would suggest that there is considerably more GT in those two mutants relative to WT. However, the levels of GT plotted in 6a are identical (not statistically different) for each of those strains.

- 6G09930 is a less bright yellow (mustard?) for GT, indicating that it produces less toxin than either 55G14390 or 6G09870. According to Figure 6a, however, the mutant produces more GT!

- 8G07280 displays a yellow shade (same as 6G09930), but in Figure 6a the mutant produces the same amount (if not less) than WT.

So, is there something wrong with this figure, or am I missing something?

Figure 7E: The bottom row of plates in 7E (i.e. the row with no visible fungus) is not described in the results or figure legend. Please do so.

Line 682: “These results suggest that KojR dependent regulation of gliT differs between GT-producing and non-producing Aspergillus species and these results are in agreement with our A. fumigatus/A. nidulans RNA-seq dataset.”

- This passage is one of many in the manuscript relating to the idea that that mechanisms of GT synthesis and resistance vary between Aspergillus species. However, the authors should discuss the possibility that there may be just as much variability within species, i.e. strain heterogeneity. Indeed, the authors mention that ∆rgdA mutants in Af293 and CEA17 display qualitatively distinct phenotypes with respect to GT production. The authors are no doubt aware (and have even reported on!) such intra-species variability with respect to other pathways/stress responses. This should be a part of this discussion.

Minor edits:

Line 372: “…produces a similar of bmGT…”

- Missing a word.

Line 421: “In contrast to the deletion of rglT, which makes all three species sensitive to GT, deletion of kojR conferred some resistance to exogenous 30 or 10 µg/ml GT, respectively…”

- The phrase “conferred some resistance” makes it sound like the ∆kojR mutants are more resistant than WT. I think the authors mean to say that the ∆kojR mutants, while still hypersensitive to GT compared to WT, are more resistant than the ∆rglT mutants.

Line 548: “…demonstrated a significantly reduced inflammation score (Figure 11C) and hyphal invasion of the tissue (Figure 11D) relative to the WT and ΔkojR::kojR strains.”

- I think the authors mean immune cell invasion, not “hyphal” invasion.

**Have all data underlying the figures and results presented in the manuscript been provided?**

Reviewer #1: Yes

Reviewer #2: Yes

Reviewer #3: Yes

PLOS authors have the option to publish the peer review history of their article (what does this mean?). If published, this will include your full peer review and any attached files.

Reviewer #1: No

Reviewer #2: No

Reviewer #3: No

---

## [Editor Report · Decision Letter 1]

4 Jan 2022

Dear Dr Goldman,

We are pleased to inform you that your manuscript entitled "Regulation of gliotoxin biosynthesis and protection in Aspergillus species" has been editorially accepted for publication in PLOS Genetics. Congratulations!

Yours sincerely,

Aaron P. Mitchell, PhD

Associate Editor

PLOS Genetics

Gregory P. Copenhaver

Editor-in-Chief

PLOS Genetics

Comments from the reviewers (if applicable):

**Data Deposition**

http://datadryad.org/submit?journalID=pgenetics&manu=PGENETICS-D-21-01568R1

**Press Queries**

---

## [Editor Report · Acceptance letter]

13 Jan 2022

PGENETICS-D-21-01568R1 

Regulation of gliotoxin biosynthesis and protection in </i>Aspergillus</i> species 

Dear Dr Goldman, 

We are pleased to inform you that your manuscript entitled "Regulation of gliotoxin biosynthesis and protection in </i>Aspergillus</i> species" has been formally accepted for publication in PLOS Genetics! Your manuscript is now with our production department and you will be notified of the publication date in due course.

With kind regards,

Zsofia Freund

PLOS Genetics

On behalf of:
